



# The Mainz Profile Algorithm (MAPA)

Steffen Beirle[1], Steffen Dörner[1], Sebastian Donner[1], Julia Remmers[1], Yang Wang[1], and Thomas Wagner[1]

[1]Max-Planck-Institut für Chemie (MPI-C), Mainz, Germany

*Correspondence to:* Steffen Beirle (steffen.beirle@mpic.de)

**Abstract.** The Mainz profile algorithm MAPA derives vertical profiles of aerosol extinction and trace gas concentrations from MAX-DOAS measurements of slant column densities under multiple elevation angles. This manuscript presents (a) a detailed description of the MAPA algorithm (v0.98), including the flagging scheme for the identification of questionable or dubious results, (b) results for the CINDI-2 campaign, and (c) sensitivity studies on the impact of a-priori assumptions such as flag
thresholds.

MAPA is based on a profile parameterization combining box profiles, which also might be lifted, and exponential profiles. The profile parameters yielding best match to the MAX-DOAS observations are derived by a Monte Carlo approach, making MAPA much faster than previous parameter-based inversion schemes, and directly providing distributions of profile parameters. The AODs derived with MAPA for the CINDI-2 campaign show good agreement to AERONET if a scaling factor of 0.8 is
applied for $O_4$, and the respective $NO_2$ and HCHO surface mixing ratios match those derived from coincident long-path DOAS measurements. MAPA results are robust to modifications of the a-priori MAPA settings within plausible limits.

## 1   Introduction

Multi AXis Differential Optical Absorption Spectroscopy (MAX-DOAS), i.e. spectral measurements of scattered sunlight under different viewing elevation angles, have become a useful tool for the determination of vertical profiles of aerosols and
various trace gases within the lower troposphere (e.g., Hönninger and Platt, 2002; Hönninger et al., 2004; Wagner et al., 2004; Wittrock et al., 2004; Clémer et al., 2010; Frieß et al., 2006), which is a key for the validation of trace gas columns derived from satellite measurements.

MAX-DOAS is based on the elevation angle dependency of spectral absorption, i.e. the differential slant column density (dSCD) determined by DOAS (Platt and Stutz, 2008). The profile retrieval is performed in two steps: first, aerosol extinction
profiles are derived based on dSCDs of the oxygen dimer $O_4$. In a second step, the concentration profiles of various trace gases detectable in the UV/vis range (such as nitrogen dioxide, $NO_2$, and formaldehyde, HCHO) can be determined.

For given aerosol/trace gas profiles, dSCDs of $O_4$ and atmospheric trace gases can be modeled by radiative transfer models (RTMs) for a sequence of elevation angles. The "profile inversion" consists of inverting this forward model, i.e. finding the extinction/concentration profile where forward modeled and measured dSCDs elevation sequences agree.

Profile inversion can be done based on a regularized matrix inversion method denoted as optimal estimation (Rodgers, 2000). It provides an elaborated mathematical framework yielding the best extinction/concentration profile estimate and the





corresponding averaging kernels for a given measurement and a-priori error (e.g., Frieß et al., 2006; Clémer et al., 2010). However, results depend on the a-priori settings, in particular the a-priori profile and its uncertainty, which are generally not known.

An alternative approach involves parameterized profiles (Irie et al., 2008; Li et al., 2010; Wagner et al., 2011; Vlemmix et al., 2011, 2015). The basic idea is to represent vertical profiles by few parameters, typically representing total column, height and shape. The profile inversion then corresponds to finding the best matching parameters. Due to the limited number of parameters, a regularization as used in optimal estimation is not required, and the method makes no a-priori assumptions on the actual profile (except that its shape can be represented by the chosen parameterization).

So far, parameter-based inversion was using non-linear least squares algorithms like Levenberg-Marquardt (LM). This is an established method; however, it has some drawbacks: First, LM is based on local linearisation, while the forward model is typically highly nonlinear in the parameters. As a consequence, the confidence intervals (CI) resulting from LM are symmetric by definition and often result in unphysical values of the fitted parameter $\pm$ CI, like a negative layer height. Second, the profile parameters are often strongly correlated, i.e. different parameter combinations can result in similar profile shapes. This implies the existence of local minima in the minimization task, making LM challenging and slowing down the inversion.

Here we present an alternative parameter-based inversion method using a Monte Carlo (MC) approach: The (finite) space of parameter combinations is covered by random numbers, and those best matching the measurement are kept. This approach directly yields distributions rather than single estimates for each parameter, thereby accounting for the correlation of parameters. In addition, the distributions do not contain unphysical parameters (as occur for LM best estimates $\pm$ CI).

The MC approach used in MAPA v0.98 is much faster than the previous LM implementation. In addition, the information on a distribution of the best matching parameters allows for a straightforward determination of the vertical concentration profiles and their uncertainties. The algorithm can also be easily adopted to additional or different profile parameterizations.

MAPA is included as representative of parameter-based algorithms in the processing chain of Fiducial Reference Measurements for Ground-Based DOAS Air-Quality Observations (FRM4DOAS), a 2-year ESA project which started in July 2016 (http://frm4doas.aeronomie.be/).

In this paper, the MAPA algorithm v0.98 is described in section 2. Exemplary results for the CINDI-2 campaign are shown in section 3. The dependency of MAPA results on a-priori settings as well as clouds is investigated in section 4. The limitations of profile inversions from MAX-DOAS measurements in general and MAPA in particular are discussed in section 5, followed by conclusions.

Table 1 lists the abbreviations and used within this study. A list of mathematical symbols used for variables and parameters is provided in Table 2.

Table 1 about here.

Table 2 about here.





## 2  Method

In this section, we describe the MAPA profile inversion algorithm. First, the measurement principle is shortly described in section 2.1. In section 2.2, the required input to the MAPA algorithm is specified. Section 2.3 describes the profile parameterization. In section 2.4, the forward model, linking profile parameters to elevation sequences of dSCDs, is provided. The profile

inversion algorithm is described in section 2.5. Section 2.6 deals with the $O_4$ scaling factor. Finally, the flagging procedure, in order to identify questionable results and outliers, is explained in section 2.7.

### 2.1  MAX-DOAS

With DOAS, slant column densities, i.e. integrated columns along the effective light path, can be determined from spectral measurements of scattered sunlight for molecules with absorption structures in the UV/Vis spectral range (Platt and Stutz,

2008). They can be converted into vertical column densities (VCDs), i.e. vertically integrated columns, by division with the so-called air mass factor.

  MAX-DOAS measurements are performed from ground based spectrometers with different elevation angles (EA) $\alpha$, including zenith sky measurements, in order to derive profile information from the EA dependency of slant column densities.

  By using the zenith measurements before and/or after a sequence of different EAs as reference spectrum within the DOAS

analysis, so-called differential slant column densities (dSCDs) $S$, representing the SCD excess compared to zenith viewing geometry, are derived. Analogously, differential air mass factors (dAMFs) $A$ relate the dSCDs $S$ to the VCD $V$:

$$V = S/A \tag{1}$$

  Note that the DOAS spectral analysis is not part of MAPA, but has to be done beforehand.

### 2.2  Input

Here we list the basic quantities needed as input for MAPA. A detailed description of the MAPA input file format is provided in the supplement.

#### 2.2.1  Viewing and solar angles

The geometry has to be specified in the MAPA input data, defined by the EA $\alpha$, the solar zenith angle (SZA) $\vartheta$, and the relative azimuth angle (RAA) $\varphi$ between viewing direction and direction of the sun. Absolute (solar and viewing) azimuth angles are

not needed.

#### 2.2.2  Elevation sequence

A sequence of $i = 1..M$ EAs with corresponding dSCDs $S_i = S(\alpha_i)$ is required for one profile to be retrieved. Below, a dSCD sequence is noted as vector $\boldsymbol{S}$, where the $i^{th}$ component corresponds to $\alpha_i$. In addition, the corresponding sequence of the DOAS fit error $\boldsymbol{S}_{\mathrm{err}}$ is required. Note that the dependency on $\alpha$ is implicit in all vectors below and not written explicitly any





more. As aerosol profiles have to be retrieved first as prerequisite for trace gas inversions, each MAPA input file must contain at least one dataset of $O_4$ dSCDs. In addition, trace gas dSCD sequences can be included as needed.

### 2.2.3 $O_4$ VCD

For the MAPA aerosol retrieval, an a-priori $O_4$ VCD $V_{O4}$ is required for each sequence in order to relate the measured
$O_4$ dSCDs to $O_4$ dAMFs (see equation 1 and section 2.4). $V_{O4}$ can be provided explicitly in the input data. If missing, it is calculated from temperature and pressure profiles. If full profile measurements are provided in the input, they are used. If ground measurements at the station are available only, they are used to construct extrapolated profiles. If no temperature and pressure information is provided in the MAPA input, ERA-interim data (Dee et al., 2011) from the European Centre for Medium-Range Weather Forecasts (ECMWF) is used for the calculation of $V_{O4}$.

## 2.3 Profile parameterization

Within MAPA, vertical profiles $p(z)$ of aerosol extinction and trace gases concentration are parameterized by 3 parameters, similar as in Wagner et al. (2011):

1. the integrated column $c$ (i.e. AOD for aerosols, VCD for trace gases),

2. the layer height $h$, and

3. the shape parameter $s \in ]0, 2[$.

A shape parameter of $s = 1$ represents a simple box profile:

$$p(z)_{c,h,s=1} = \begin{cases} c/h & \text{for } z \leq h \\ 0 & \text{for } z > h \end{cases} \tag{2}$$

For a shape parameter of $0 < s < 1$, the fraction $s$ of the total column $c$ is placed within a box. The remaining fraction $(1-s)$ is exponentially declining with altitude:

$$p(z)_{c,h,s<1} = \begin{cases} s \times c/h & \text{for } z \leq h \\ s \times c/h \times \exp(-\frac{z-h}{h} \times \frac{s}{1-s}) & \text{for } z > h \end{cases} \tag{3}$$

A shape parameter of $2 > s > 1$ represents an elevated layer from $h_1$ to $h$ of thickness $h_2$:

$$p(z)_{c,h,s>1} = \begin{cases} 0 & \text{for } z < h_1 \\ c/h_2 & \text{for } h_1 < z \leq h \\ 0 & \text{for } z > h \end{cases} \tag{4}$$



with

$$h_1 = (s-1)h$$
$$h_2 = (2-s)h \tag{5}$$
$$h_1 + h_2 = h$$

Equations 3 and 4 converge to a box profile for $s \rightarrow 1$, thus equations 2 to 4 describe a set of parameterized profiles which are continuous in $s$. Figure 1 exemplarily displays extinction profiles for $c$=1 and different heights $h$ and shape parameters $s$.

Figure 1 about here.

Alternative parameterizations (like a linear increase from ground to $h$ (compare Wagner et al., 2011), or even completely different profile shapes) might be used instead or in addition in future MAPA versions. This would require the calculation of corresponding look-up tables (LUTs) for dAMFs (see below).

## 2.4 Forward model

In this section the forward model (fm) is specified which connects the profile parameters $c$, $h$, and $s$, with dSCDs for the given solar and viewing geometry specified by $\vartheta$, $\varphi$, and $\alpha$.

Essentially, the forward model is given by eq. 1: $S = V \times A$, where the dAMF depends on profile parameters and solar and viewing geometry. Within MAPA, dAMFs have been calculated offline with the radiative transfer model McArtim (Deutschmann et al., 2011) for fixed nodes for each parameter, and stored as look-up-table (LUT). Within MAPA profile in-

version, these multi-dimensional LUTs are interpolated linearly for the given parameter values. For details on the dAMF LUT properties see Appendix A.

Note that the profile parameterization (sec. 2.3) is the same for aerosols and trace gases. The forward models for aerosols and trace gases, however, are similar (and the profile retrieval is based on the same code as far as possible), but not identical. This is due to the fact that the column parameters $c_{aer}$ and $c_{tg}$ have different meanings in the context of $\boldsymbol{S}$ and $V$: For aerosols,

$c$ equals the AOD $\tau$, which is completely independent from the $O_4$ VCD. For trace gases, $c$ equals the VCD $V_{tg}$.

Below, the forward models will be described for both $O_4$, which is the basis for retrieving aerosol profiles, and trace gases.

### 2.4.1 Forward model for aerosols

For aerosols, the $O_4$ dAMF is a direct function of the profile parameters $c_{aer} (\equiv \tau), h_{aer}, s_{aer}$ and viewing geometry $\vartheta, \varphi$:

$$\boldsymbol{A}^{O4} = f(c_{aer}, h_{aer}, s_{aer})|_{\vartheta, \varphi} \tag{6}$$

The corresponding dSCD is:

$$\boldsymbol{S}_{fm}^{O4} = V_{apriori}^{O4} \times \boldsymbol{A}^{O4} \tag{7}$$

The respective VCD of $O_4$ (or vertical profiles of pressure and temperature, which allow for the calculation of $V_{O4}$) has to be provided in the MAPA input or is calculated from ECMWF profiles.



### 2.4.2 Forward model for trace gases

For trace gases, the dAMFs also depend on the aerosol profile parameters as determined from the analysis of $O_4$ dSCDs[1], but not on the trace gas VCD $c_{tg}$, as long as optical depths are low (which is a prerequisite for DOAS analysis):

$$\boldsymbol{A}^{tg} = f(h_{tg}, s_{tg})|_{\vartheta, \varphi, c_{aer}, h_{aer}, s_{aer}} \tag{8}$$

The corresponding dSCD is:

$$\boldsymbol{S}_{fm}^{tg} = V^{tg} \times \boldsymbol{A}^{tg} = c_{tg} \times \boldsymbol{A}^{tg} \tag{9}$$

The trace gas VCD $V^{tg}$ is identical to the column parameter $c_{tg}$.

### 2.5 Profile inversion

The forward model as defined above translates the aerosol and trace gas profile parameters $c$, $h$ and $s$ into dSCD sequences $\boldsymbol{S}_{fm}$. Within profile inversion, the task is now to find those model parameters yielding the "best match" (bm) between $\boldsymbol{S}_{fm}$ and the measured dSCD sequence $\boldsymbol{S}_{ms}$. Typically, "best match" is defined in terms of least-squares of the residue, i.e. the root-mean-square (RMS)

$$R = \sqrt{\frac{(\boldsymbol{S}_{fm} - \boldsymbol{S}_{ms})^2}{M}} \tag{10}$$

is minimized, with $M$ being the number of EAs (i.e. the length of $\boldsymbol{S}$).

In previous parameter-based inversion schemes, the best matching parameters have been determined by non-linear least squares algorithms like Levenberg-Marquardt (Li et al., 2010; Wagner et al., 2011; Vlemmix et al., 2015). This approach, however, has some drawbacks, in particular

- as the parameters are highly correlated and local minima can exist, high computational effort, i.e. multiple minimization calls with different initial values, is needed in order to soundly determine the absolute minimum.

- as the least-squares algorithms are based on local linearisation, the resulting parameter uncertainties are per construction symmetric. The resulting parameter range spanned by the fitted parameter $\pm$ CI is often unphysical (e.g. $h < 0$ or $s > 2$) and thus meaningless.

Within MAPA (from v0.6 onwards), thus a different, Monte-Carlo (MC) based approach is chosen. The idea is to (a) generate multiple random sets of profile parameters, (b) calculate the respective dSCD and RMS, and (c) keep those yielding the best agreement. This approach results in a best matching set of parameters, plus an ensemble of parameter sets with similar low $R$, which reflects the uncertainty range of the estimated profile parameters, which per construction only contains physically valid values.

Section 2.5.2 describes the details of the MC inversion approach, which is used for the determination of $h$, $s$, and $c_{aer}$. Before that, in section 2.5.1 the determination of $c_{tg}$ is described which is implemented differently by a simple linear fit.

---

[1]Note that it is not possible to directly use an a-priori vertical aerosol extinction profile within MAPA trace gas inversion.





### 2.5.1 VCD: linear fit

The dSCD forward model is highly non-linear in $h$ and $s$ and also in AOD $c_{aer}$. These parameters are derived by MC as described in detail the next section.

The trace gas VCD $c_{tg}$, on the other hand, is just a scaling factor of $\boldsymbol{A}$ (eq. 9). Thus, for a given set of profile parameters, and a given sequence of measured dSCDs, the best matching trace gas VCD $c_{tg} = V_{bm}$ can just be determined by a linear fit (forced through origin) of $V$:

$$V_{bm} = \frac{\boldsymbol{S}_{ms} \cdot \boldsymbol{A}}{\boldsymbol{A} \cdot \boldsymbol{A}} \tag{11}$$

(Note that $\boldsymbol{S}$ and $\boldsymbol{A}$ are vectors, and the multiplications are scalar products).

In other words, the best matching $V$ equals the mean of $V_i$ for individual elevation angles, weighted by the respective dAMF (i.e., sensitivity).

The same formalism is used to define a VCD uncertainty $V_{err}$ as the weighted mean of dSCD errors (from DOAS analysis) for individual EAs. $V_{err}$ is used as column error proxy within the flagging algorithm in order to decide if the found variability of column parameters is within expectation or not (see section 2.7 for details).

### 2.5.2 Other profile parameters: Monte-Carlo

Within MAPA, profile parameters are determined by just covering the parameter space by random numbers[2] and keeping the matches. In detail, the following steps are performed:

1. limits are defined for each parameter[3],

2. $n_{tot}$ sets of random parameters are drawn[4,5],

3. the RMS $R$ is calculated for each random parameter set,

4. the lowest RMS is identified as "best match" (bm) $R_{bm}$, and

5. an ensemble of up to $n_{sel}$ parameter sets with $R/R_{bm} < F$ is kept.

Table 3 lists the default values for parameter limits, number of randoms, and thresholds for MAPA v0.98. The impact of variations of these settings is discussed in section 4.1.

The steps listed above are iterated 3 times, where the resulting ensemble is used to narrow down the parameter limits for the next iteration. I.e., if lowest $R$ is always found for low $s$, the limits for $s$ will be narrowed for the next iteration. As the total number of randoms stays the same, this procedure results in increasingly finer spacing of random numbers.

---

[2]MAPA also provides the option to fix each of the parameters to a predefined value.

[3]This approach (as well as the implementation the dAMF as a LUT) is only possible since the (physical or plausible) parameter ranges are limited.

[4]By default the random number generator is initialized with a seed $\beta$ in order to generate reproducible results

[5]Parameter combinations yielding thin elevated layers (less than 50 m thick), which correspond to high $s$ and low $h$, are excluded, as the respective profiles might not be vertically resolved within the RTM calculation of the dAMF LUT.





The procedure results in a best matching parameter set, plus an ensemble of acceptable parameter sets. For each parameter set, also the corresponding VCD $V_{bm}$ is determined by eq. 11.

Table 3 about here.

### 2.5.3 Best match and ensemble statistics

MAPA yields the best matching parameter combination. The corresponding vertical profile is given by equations 2-4. In addition, MAPA yields an ensemble of parameter sets with similar agreement between measurement and forward model. From this ensemble, the following statistics are derived for both the profile parameters as well as the corresponding vertical profiles:

- mean and standard deviation (weighted by $1/R^2$),

- 25 and 75 percentiles, and

- absolute minimum and maximum.

The mean profiles are often smeared out; in particular strong vertical gradients (occurring for $s \geq 1$) are smeared. The degree of smearing depends on the variability of parameters within the ensemble, which is determined by $R_{bm}$ and the a-priori threshold for accepted RMS values $F$.

Note that mean±standard deviation might exceed pyhsical limits for parameters and profiles, similar to LM fit results $\pm$ CI. The 25/75 percentiles avoid this. Only for $c_{tg}$, which is not determined by MC but by a linear fit, unphysical (negative) VCDs and concentrations can occur. These can be understood as noise for quasi-zero VCDs, and must not be set to 0 or skipped in order to keep unbiased means.

Below we mainly focus on the best match (bm) and weighted mean (wm) of parameters and profiles.

Within trace gas retrievals, aerosol profile parameters are required for accessing the dAMF LUT. For this, the best matching parameters are taken. Due to nonlinearities (the mean of ensemble profiles does not equal the profile corresponding to the mean parameters), it is not possible to take mean parameters for this. If one is interested in the actual aerosol profile and its uncertainty, however, the mean profile and the percentiles might still yield valuable information.

Figure 2 exemplarily displays $O_4$ dSCDs (top) and the retrieved aerosol extinction profiles (bottom) for an afternoon sequence on 15 (left) and 23 (right) September 2016. Best match, weighted mean, 25/75 percentiles and min/max are shown. For these examples, a scaling factor of 0.8 has been applied for $O_4$ (see next section). This choice will be justified in section 3.

Figure 3 displays the respective dSCDs and profiles for $NO_2$.

Figure 2 about here.

Figure 3 about here.





## 2.6 Scaling of $O_4$ dSCDs

Some previous studies have reported on a significant mismatch between modeled and measured dSCDs of $O_4$, which is usually accounted for by applying an empirical scaling factor (SF) $f$ of about 0.8 to the $O_4$ dSCDs, while other studies (e.g. Ortega et al., 2016) do not see a need for a SF, for reasons still not understood. An in-depth discussion of the $O_4$ SF is provided in Wagner et al. (2018).

MAPA provides the option for defining a fixed a-priori scaling factor $f$ of e.g. 0.8. Note that within MAPA, the measured dSCD is unchanged (in order to have the same measured dSCD in plots and result files for comparison), but the modeled dSCD is divided by $f$ instead.

Another option arises from the profile inversion procedure: the linear fit of the best matching VCD (eq. 11), used for the determination of $c_{tg}$, can likewise be used to determine the best matching VCD of $O_4$. This defines the best matching SF as

$$f_{bm} = V_{apriori}/V_{bm} \tag{12}$$

Note that extreme deviations of $f$ from 1 are flagged later (see section 2.7).

As the issue of the $O_4$ SF is still not understood and its value or even its need is highly debated within the community, it was decided to always run MAPA with 3 different settings for $f$ within the FRM4DOAS project:

1. no scaling of $O_4$ dSCDs, i.e. $f \equiv 1$,

2. a SF of $f = 0.8$,

3. a variable (best matching) SF $f_{bm}$.

This setup has also been adopted as default in MAPA v0.98. The comparison of the MAPA results for the different settings for $f$ for different campaigns, instruments, and conditions hopefully will help to clarify the SF issue in the future.

## 2.7 Flags

The profile inversion scheme as described in section 2.5 just searches for the parameter combinations yielding best agreement in terms of lowest $R$. Thus, it will always result in a "best match", even if the agreement between measured and modeled dSCDs is actually poor, or the resulting parameter ensembles are inconsistent. Therefore, additional information is needed in order to evaluate whether the resulting profile is trustable or not.

Within MAPA, flags raising warnings or errors are provided based on the performance of the profile inversion. Note that output is generated for each elevation sequence, also for those flagged by an error, and the final decision on which profiles are considered as meaningful is in the users hand. Nevertheless, we strongly recommend to consider the raised warnings and errors; error flags should generally lead to a rejection of the affected profiles.

In this section we describe the warning and error flag criteria and thresholds for MAPA v0.98. The thresholds, denoted by $\Theta$ below, are defined in the flag configuration file and can easily be modified. However, any change should only be made for good reasons and has to be tested carefully.



Within the FRM4DOAS processing chain, MAPA has to provide reasonable output for a wide variety of instruments and measurement conditions, which could not all be tested beforehand. Thus, the general strategy is to have low thresholds for warnings (conservative approach), and higher thresholds for errors, indicating cases which do not make sense at all.

The flags defined in MAPA v0.98 can be grouped in 4 categories:

1. Flags based on the agreement between forward-modelled and measured $\boldsymbol{S}$,

   2. Flags based on consistency of the ensemble of derived MC parameters,

   3. Flags based on the profile shape, and

   4. Miscellaneous.

Below the different flag criteria are explained in detail. The default warning and error thresholds for MAPA v0.98 are listed in
table 4.

Table 4 about here.

### 2.7.1   RMS

The RMS $R$ as defined in eq. 10 reflects the agreement between measured and best matching $\boldsymbol{S}$. Thus $R$ might directly be used for flagging, as high RMS values generally indicate that the forward model is not capable of reproducing the measurement. In
order to account for the instrument dependent uncertainty of the measured dSCDs, the flag threshold $\Theta_R$ is given in units of the typical (sequence median) DOAS fit error $S_{\mathrm{err}}$.

Since $S$ scales with the actual VCD $V$ and the dAMF $\boldsymbol{A}$, $R$ is generally large for high trace gas columns and/or high dAMFs. The first corresponds to polluted episodes, while the second represents conditions under which the MAX-DOAS technique is particularly sensitive. Both cases are of particular interest, but would often be flagged if just a threshold for $R$ based on typical
values is defined.

Thus we also consider the RMS normalized by the maximum dSCD $S_{\mathrm{max}}$:

$$R_{\mathrm{n}} = R/S_{\mathrm{max}} \tag{13}$$

Due to the normalization, $R_{\mathrm{n}}$ removes the scaling of $R$ with $V$ and $\boldsymbol{A}$. However, for very low $V$ or $\boldsymbol{A}$, i.e. dSCDs about 0, $R_{\mathrm{n}}$ can become quite large and the intrinsic noise of the dAMF LUT (if calculated by MC RTM as McArtim) matters.
Warning and errors are thus only risen if the values for $R$ and $R_{\mathrm{n}}$ both exceed the thresholds given in table 4.

### 2.7.2   Consistency

In addition to the best matching parameters, MAPA derives an ensemble of parameter sets yielding similar agreement in terms of $R$. But this does not mean that the ensemble parameters are consistent. While different height and shape parameters might





be acceptable (and just result in a larger profile uncertainty), the column parameter is an important integrated property of the profile. Thus a consistency flag is defined based on the spread of the column parameter within the ensemble.

In order to evaluate if the spread is acceptable or not, we define $\varepsilon$ as proxy of the column uncertainty. For aerosols, $\varepsilon$ is defined in absolute terms in the MAPA flag configuration (default: 0.05). For trace gases, $\varepsilon$ is set to $V_{\mathrm{err}}$, which is derived from the SCD error $S_{\mathrm{err}}$ provided in the input data according to eq. 11.

Based on $\varepsilon$, we define the tolerated deviation for $c$ as

$$c_{\mathrm{tol}} = \Theta_{\mathrm{abs}} \times \varepsilon + \Theta_{\mathrm{rel}} \times c_{\mathrm{bm}}, \tag{14}$$

consisting of an absolute and a relative term. I.e., for low columns, the tolerance is dominated by $\varepsilon$ scaled with the absolute threshold defined in the flag settings, whereas for high columns, the relative term $\Theta_{\mathrm{rel}} \times c_{\mathrm{bm}}$ dominates.

Flags are raised if the ensemble standard deviation of $c$ or the difference between $c_{\mathrm{bm}}$ and $c_{\mathrm{wm}}$ exceed the column tolerance.

The consistency flag indicates that the observations have been reproduced with comparable RMS by parameter sets with considerably different column parameters. I.e., the dSCD sequence shows no strong dependency on $c$, and MAXDOAS measurements are thus not sensitive for $c$ under these conditions.

### 2.7.3 Profile shape

MAXDOAS measurements are sensitive to the lower troposphere up to about 2-3 km (Frieß et al., 2006). Profiles reaching up in the free troposphere thus have to be treated with care. Within MAPA v0.98, these cases are identified and flagged based on two quantities:

- the fitted height parameter $h$, and

- the integrated profile within the lower troposphere $c_{\mathrm{LT}}$ (default: below 4km).

A flag is raised if $h > \Theta_h$ or $c_{\mathrm{LT}}/c_{\mathrm{bm}} < \Theta_{\mathrm{LT}}$, but only if also the column $c_{\mathrm{bm}}$ exceeds the column detection limit

$$c_{\mathrm{DL}} = \Theta_{\mathrm{DL}} \times \varepsilon, \tag{15}$$

as for very low columns, the profile shape can not be specified anyhow. Note that per default $\Theta_{\mathrm{abs}}$ equals $\Theta_{\mathrm{DL}}$, thus $c_{\mathrm{DL}}$ is the same as the absolute tolerance term in equation 14, but MAPA also allows to have different thresholds for both.

### 2.7.4 Miscellaneous

In addition, the following flags are defined:

- Missing elevation angles:

  In case of incomplete elevation sequences, an error is raised during the MAPA preprocessing. As profile inversion determines 2–3 parameters for about 2–4 degrees of freedom (Frieß et al., 2006), the number $M$ of available EAs must not be too small, otherwise (default: $M < 5$) an error is raised.





Note that for the results for CINDI-2 shown in the following sections, all incomplete sequences are removed first, as this is related to missing input data, not to the MAPA performance.

– NaNs:

Best match, mean and std of $c$ are checked for NaNs. These might occur in case of NaNs present in the input data. NaN

values automatically raise an error.

– AOD:

High AOD likely indicates the presence of clouds. But even in case of cloud free conditions, high AOD indicate complex ratiative transfer conditions. Thus flags are raised if $c_{aer} \equiv \tau > \Theta_\tau$.

– RAA:

If the relative azimuth angle is too low ($\varphi < \Theta_\varphi$), i.e. the instrument is directed towards the sun, and the AOD is high enough ($c_{aer} \equiv \tau > \Theta_{\varphi,\tau}$), a warning flag is raised, as for this scenario the forward peak of aerosol scattering matters, which is only roughly captured by the Henyey-Greenstein parameterization used in RTM.

– $O_4$ scaling factor:

MAPA provides the option to derive a best matching SF for $O_4$ (see section 2.6). Large deviations of the SF from 1 are

flagged according to the thresholds defined in table 4.

### 2.7.5 Cloud flag

Several studies have characterized cloud conditions based on MAX-DOAS elevation sequences, making use of radiance and color index and their (inter- and intra sequence) variability (Gielen et al., 2014; Wagner et al., 2014, 2016; Wang et al., 2015). While dedicated algorithms have been optimized for specific instruments, it is difficult to automatize these algorithms as MAX-

DOAS instruments are usually not radiometrically calibrated. I.e. the thresholds for cloud classification have to be adjusted for each instrument.

Therefore, no automatized cloud flagging algorithm is included within MAPA so far. However, MAPA provides the option to add external cloud flags to the MAPA input. A-priori flags in input data are treated like the other flags during MAPA processing, included in the calculation of the total flag (see below), and written to the MAPA output.

Similarly, also other external flags (like an "instrument failure flag" etc.) can easily be added.

In section 4.5, we investigate the impact of clouds on MAPA results and how far the current default settings for MAPA flagging succeed in identifying cloudy scenes.

### 2.7.6 Total flag

As final step of the flagging procedure, a total warning or error flag is raised if any of the flags defined above indicate a warning

or an error, respectively.





## 3 Results

In this section we present MAPA results exemplarily for dSCD sequences of $O_4$, $NO_2$ and HCHO measured during the Second Cabauw Intercomparison campaign of Nitrogen Dioxide Measuring Instruments (CINDI-2) during September 2016 (Kreher et al., in prep.). We focus on two days, September 15 and September 23, which are mostly cloud free and have also been selected

as reference days within CINDI–2 intercomparisons (Tirpitz et al., in prep.). The required $O_4$ VCD is derived from ECMWF interim temperature and pressure profiles, interpolated in space and time.

For details on the MPIC MAX-DOAS instrument and DOAS fit settings see the supplementary material provided by Kreher et al. (in prep.).

### 3.1 Aerosols

$O_4$ dSCDs have been analyzed according to the DOAS settings specified in table A3 in Kreher et al. (in prep.), but with sequential instead of noon reference spectra. Fig. 4 displays the MAPA results based on the original $O_4$ dSCD sequences. In subplots (a) and (b), the valid vertical extinction profiles are displayed for the two selected days. The invalid sequences are marked by the respective flags (symbols as in (c)). In (d) and (e), the respective timeseries of AOD are shown and compared to AERONET measurements (Dubovik and King, 2000)[6]. In (c), flag statistics are provided for all available measurements

during the campaign, covering the period from 9 September to 2 October 2016. Panel (f) displays a scatterplot of MAPA AOD compared to 15 minute AERONET means where available for the full campaign. Note that the scales are not linear in order to cover the different order of magnitude in AOD for the two selected days.

Figure 4 about here.

A large fraction of sequences is flagged (overall, less than 1/4 of all sequences are valid). On 23 September, not a single

valid sequence was found from 9:00 to 14:00. Even worse, the remaining AODs do not match AERONET (e.g. afternoon of 23 September).

This poor performance is related to a general mismatch between modeled and measured dSCDs, as has been also found for other campaigns in the past (see Wagner et al., 2018, and references therein). We thus perform another MAPA retrieval with an $O_4$ SF of $f = 0.8$ (Fig. 5).

Figure 5 about here.

The application of a SF largely improves MAPA performance and the agreement to AERONET. A far higher number of sequences is now categorized as valid. The temporal pattern of AOD generally matches well between MAPA and AERONET. Correlation to AERONET AOD (15 minute averages) is as good as $r = 0.874$ with a mean deviation of $0.012 \pm 0.067$.

Fig. 6 displays MAPA results based on a variable SF. They are overall similar to the results for a fixed SF of 0.8. For the

complete campaign, mean and std of the best matching SF in variable mode are $0.85 \pm 0.08$.

---

[6]The original level 2 AERONET AOD determined at 440 nm has been transferred to 360 nm by assuming an Ångström exponent of 1





Figure 6 about here.

Below we focus on the results for $f = 0.8$ as this is clearly defined, whereas the free scaling factor increases the degrees of freedom, and different effects might affect the best matching SF.

### 3.2 Nitrogen Dioxide (NO$_2$)

5    The MPIC DOAS retrieval for NO$_2$ has been performed in a fit window slightly different from that of O$_4$, i.e. 352 to 387 nm. Figure 7 displays MAPA results for NO$_2$. The bottom row now displays the mixing ratio in the lowest 200 m layer instead of the total column. For comparison, mixing ratios derived from long path (LP) DOAS measurements are shown. The LP measurements have been provided by Stefan Schmitt (IUP Heidelberg). Details on LP instruments and retrieval are given in Pöhler et al. (2010) and Eger et al. (in prep.).

10    NO$_2$ profiles are generally far closer to the ground compared to aerosol profiles, which is expected, as sources are located at the ground and the NO$_x$ lifetime of some hours is far shorter than that of aerosols.

Comparison of the NO$_2$ mixing ratio in the lowest 200m layer to LP measurements yields a correlation of $r = 0.887$. The mean difference between MAPA and LP mixing ratios for valid sequences is 0.84±2.26 ppb.

The flagging is strongly dominated by the aerosol flag inherited from the aerosol analysis.

15                               Figure 7 about here.

### 3.3 Formaldehyde (HCHO)

HCHO dSCDs have been analyzed according to the DOAS settings specified in table A4 in Kreher et al. (in prep.), but with a sequential instead of a noon reference spectrum.

Figure 8 displays MAPA results for HCHO. Profiles reach up higher than for NO$_2$ as expected due to HCHO being secondary

20 product in VOC oxidation.

As for NO$_2$, the flagging is dominated by the aerosol flag. But in addition, several more sequences are flagged, with contributions from all RMS, consistency and profile shape flags.

Comparison of the HCHO mixing ratio in the lowest 200 m layer to LP measurements yields a correlation of $r = 0.937$. The mean difference between MAPA and LP mixing ratios for valid sequences is 0.35±0.56 ppb.

25                               Figure 8 about here.

## 4   Sensitivity studies

The MAPA profile inversion and flagging algorithms are controlled by a-priori parameters. These have been defined by plausible assumptions. In this section we investigate how sensitive the MAPA results are for different a-priori settings, based on the aerosol retrieval for CINDI-2 applying a fixed SF of 0.8, and its comparison to AERONET.





In section 4.1, the sensitivity on MC settings is investigated. The impact of flagging thresholds is analyzed in section 4.2. Note that flag settings can easily be modified a-posteriori, while different MC settings require a complete reanalysis. Table 5 lists the investigated variations for both MC and flag settings, and the impact on the number of valid sequences and the resulting AOD, as compared to AERONET. It also includes results for a previous MAPA version as well as for different $O_4$ SF, as discussed in sections 4.3 and 4.4.

Finally, section 4.5 investigates the dependency of MAPA flag statistics on cloud conditions.

Table 5 about here.

## 4.1 MC settings

In this section, the MC settings as defined in the MAPA MC configuration file are modified one by one.

## A Random seed

The random generator can be initialized by the seed $\beta$ provided in MAPA MC configuration. This allows to generate reproducible results even though the method is based on MC. We have tested two alternative seed values just to check how strong the impact of usage of random numbers is. The number of valid sequences and the results for AOD only change slightly for different random sets.

## B Number of randoms

As default, each profile parameter is sampled by $a$=50 values per variable. I.e. for the height parameter, which is within 0.02 and 5 km, the average spacing of the raster in $h$ dimension is about 0.01 km (note that the average spacing gets smaller in the second and third iteration of the narrowed parameter intervals, see section 2.5.2). The total number of random parameter sets $n_{\text{tot}}$ is $a$ to the power of MC variables, i.e. $50^3$=125000 for aerosols. This corresponds to a duration of about 3 seconds per elevation sequence on a normal PC.

If $a$ is lowered to 20 ($n_{\text{tot}}$=8000), the profile inversion is much faster. But only 269 instead of 324 sequences are identified as valid. However, the remaining profiles show good agreement to AERONET. If a number of $a$=100 ($n_{\text{tot}}$=$10^6$) is chosen, about 20 more sequences are labeled as valid compared to the baseline. But the agreement to AERONET gets slightly worse, and the required time is more than 10 fold.

The impact of $a$ on the number of valid sequences can be understood as for higher $a$, the parameter space is sampled on finer resolution. Thus the RMS of the best match, $R_{\text{bm}}$, generally becomes lower. Consequently, the parameter ensemble defined by $R < F \times R_{\text{bm}}$ is more homogeneous, and less sequences are flagged as inconsistent.

We found $a$=50 as good compromise between computation time and the number of valid sequences.




## C   Ensemble threshold for RMS

MAPA determines the best matching parameter combination by the lowest RMS $R$. In addition, an ensemble of parameter sets is kept with $R < F \times R_{\min}$. The resulting ensemble allows to estimate the uncertainty of the derived parameters and profiles. Per default, $F$ is set to 1.3. We have tested smaller and higher values for $F$ in scenarios C1 and C2.

For a low value of $F = 1.1$, a far higher number of sequences is characterised as valid. This is due to the variety of parameters in the ensemble is being lowered, and consequently the consistency thresholds are less often exceeded. Another side effect is that also the profile uncertainty estimate, which is derived from the variability of profile parameters, is lowered. For the extreme scenario $F_R \rightarrow 1$, only the best matching parameter set would be left, which would be close to the result from LM if the number of randoms is high enough. Interestingly, the agreement to AERONET is slightly worse for a low $F$.

Contrary, a higher value for $F$ results in less valid sequences (as more sequences are characterized as inconsistent), but the remaining ones show better agreement to AERONET.

## D   Shape parameter limits

The shape parameter $s$ determines the profile shape according to sec. 2.3. Modifying the allowed parameter range thus changes the basic population of possible profile shapes within the random ensemble.

As default, the shape parameter almost covers the nodes of the dAMF LUT, except for $s_{\min}$ which is set to 0.2. Changing this to 0.1 means allowing for boxes with long exponential tails, which are likely flagged later by the profile shape flag due to the LT criterium. Setting $s_{\min}$=0.1 worsens the performance (less valid sequences as expected, slightly poorer agreement to AERONET), while a value of 0.5 improves the difference, but not the correlation to AERONET.

Setting $s_{\max}$ to 1.5 (i.e. removing very thin elevated layers from the basic population) has almost no effect on the CINDI-2
aerosol results.

### 4.2   Flag settings

Here we modify the flag settings and thresholds as defined in the MAPA flag configuration file one by one. Except for the thresholds for height parameter and AOD, the default values are halved and doubled.

### a   RMS

We have changed the RMS thresholds for $R$ and $R_n$ in both directions. A change of the threshold of $R$ has hardly any effect in the case of our CINDI-2 results. This might of course be different for other instruments or measurement conditions.

Lowering the threshold for $R_n$ has a tremendous effect: 86 more sequences would be flagged compared to the default. The remaining sequences show a better correlation, but slightly worse agreement to AERONET AOD.





### b Column uncertainty proxy

For trace gases, $\varepsilon_{tg}$ can be determined from the dSCD sequence (see sect. 2.5.1). This is not possible for the aerosol retrieval. Instead, $\varepsilon_\tau$ has to be defined by the user.

Per default, $\varepsilon_\tau$ is set to 0.05. A lower/higher value for $\varepsilon_\tau$ slightly decreases/increases the number of valid sequence, but the
agreement to AERONET does hardly change.

### c Consistency

The variations of the thresholds related to the consistency flag can be summarized as follows: More strict criteria (c1&c3) result in less valid sequences, but a slightly better agreement to AERONET. Vice versa, less strict criteria (c2&c4) result in more valid sequences with poorer agreement to AERONET. We consider the current default settings as plausible and a good
compromise.

### d Profile shape

Here we focus of variations of $\Theta_h$. The impact of modifications of $\Theta_{LT}$ (not shown) is similar.

If $h_{max}$ is set to 4 km, which was the default value in previous MAPA versions (compare section 4.3), more sequences are labeled as valid, but the agreement to AERONET gets worse. For instance, for the measurements around 16:00 on 15
September, where MAPA AOD is far higher than AERONET, a warning was raised by the height parameter (see Fig. 5 (a) and (d)). For $h_{max}$=4 km, these sequences are labeled as valid.

If the threshold for $h_{max}$ is lowered to 2 km, less valid sequences remain, but those show significantly better agreement to AERONET, both for correlation and difference. This reflects that MAX-DOAS measurements are mainly sensitive for profiles close to the ground (Frieß et al., 2006). Consequently, inversion results for profiles reaching up to higher altitudes have higher
uncertainties.

### e AOD

Modifications of the AOD threshold have almost no effect. This might however be different for measurements under higher aerosol load.

### 4.3 MAPA version 0.96

In table 5, also the results for previous MAPA version 0.96 are included. This version was used for the FRM4DOAS verification study (Richter and Tirpitz, in prep.).

Versions 0.96 was based on the same MC algorithm with the same MC settings as v0.98. However, the flag definitions and thresholds differ slightly. The main difference is that the height threshold for the profile shape flag was set to 4 km in v0.96. Consequently, v0.96 results in more valid sequences, but with slightly poorer agreement to AERONET AOD, similar as for
variation d2.



## 4.4   Different scaling factors

The results presented above are based on an $O_4$ SF of 0.8. If instead no scaling factor would be applied, a far higher number of sequences would be flagged, and only 218 sequences remain. These show a good correlation to AERONET, but a systematic bias of -0.115 (compare Fig. 4). The ratio of the mean AOD from MAPA vs. AERONET is 0.53, i.e. MAPA results are too low by a factor of 2 on average if no SF is applied.

5        If the SF is considered as variable, about 30 more sequences are valid, with similar agreement to AERONET as for a fixed SF of 0.8.

## 4.5   Clouds

As MAX-DOAS measurements are usually not radiometrically calibrated, a cloud classification cannot easily be automatized.
Thus, so far no dedicated cloud flag is included in MAPA default settings. In this section we investigate how far MAPA flags and results for aerosol retrieval depend on cloud conditions, and how far the current MAPA flags are able to catch clouded conditions.

We have derived a cloud classification based on the scheme described in Wagner et al. (2016), with thresholds adjusted for CINDI-2. Note that cloud information is missing for some elevation sequences due to missing $O_4$ dSCDs for single elevation
angles.

Fig. 9 displays the classification of clouds during CINDI-2 for all elevation sequences as well as for those sequences where AERONET AOD measurements are available. During the campaign, 33% of the sequence are categorized as cloud free. If only sequences with coincident AERONET measurements are considered, 72% are cloud free, and the remaining cases are to equal parts cloud hole conditions or missing cloud information. Only 2% are characterized as broken cloud, and no sequence
as continous cloud. Thus, a comparison of MAPA results to AERONET to large extent implies a cloud filtering even if no dedicated cloud flag is available.

Figure 9 about here.

Figure 10 about here.

We have investigated the MAPA flag statistics for different cloud conditions in Fig. 10. For the full campaign, 36% of
all sequences are valid. If only cloud free scenes with low aerosol are considered, 68% are valid, while for clouded scenes (broken+continuous clouds), only 13% are valid. Note that the flags for RMS, consistency, height and AOD all contribute significantly to the flagging of clouded scenes.

For the selection of sequences where AERONET is available, 65% sequences are valid.

As demonstrated for CINDI–2, most clouded cases are successfully flagged in MAPA. But a significant number of cloud
hole/broken cloud scenes still remains. We thus recommend that the user applies an additional cloud classification according to e.g. Wagner et al. (2016), and to flag cloud holes with a warning, and continuous and broken cloud scenes with an error.





## 5    Limitations

In this section we discuss challenges and limitations of MAX-DOAS profile inversion, which have to be kept in mind when interpreting the results and comparing them to other datasets. We start with issues generally affecting MAX-DOAS inversions, followed by MAPA specific issues.

### 5.1    General limitations of MAX-DOAS profile inversions

In this section we discuss general MAX-DOAS limitations, which also account for optimal estimation algorithms. Still, the issues are discussed from a MAPA perspective.

#### 5.1.1    RTM assumptions

Within forward models, RTM calculations are required which need a-priori information on e.g. aerosol properties like single
scattering albedo. If this information is not available and wrong assumptions are made, resulting profiles are biased.

For MAPA, the dAMF LUT used in the forward model has been calculated based on a-priori assumptions as specified in Appendix A. Currently, additional LUTs for different a-priori settings are calculated which might be used alternatively in future and allow to quantify the impact of a-priori RTM assumptions on MAPA results.

#### 5.1.2    Horizontal gradients

Current MAX-DOAS inversion schemes are based on the assumption of horizontally homogeneous layering. In reality, however, aerosol and trace gas distributions reveal horizontal gradients, as can be clearly demonstrated by comparing the results for different azimuthal viewing directions (e.g. Wagner et al., 2011).

It is very challenging to account for horizontal gradients in trace gas inversion algorithms, as (a) the degrees of freedom are numerous (and have to be limited by some simplifications), and (b) fully 3D radiative transfer modelling has to be performed,
which is only supported by few RTMs (e.g. McArtim), and far more time consuming.

Currently, an MAX-DOAS inversion scheme accounting for horizontal gradients is developed at MPIC (Remmers et al., in prep.) based on simultaneous measurements in four azimuth directions. For MAPA, horizontal gradients are so far ignored, but corrections might possibly be added in future versions based on the lessons learned in Remmers et al. (in prep.).

#### 5.1.3    Clouds

Clouds are usually ignored in MAX-DOAS inversion. Thus, elevation sequences affected by clouds have to be flagged. Several algorithms have been proposed for the classification of cloud conditions from MAX-DOAS meaurements (Gielen et al., 2014; Wagner et al., 2014, 2016; Wang et al., 2015), using the zenith values as well as EA dependency of radiances and color indices. However, as MAX-DOAS radiances are usually not calibrated, it is not straightforward to define a universal standardized cloud classification for all kind of instruments. Instead, thresholds have to be adjusted for each instrument.
For CINDI–2, the MAPA flagging scheme raises a warning or error in 87% of all clouded scenes.



### 5.1.4 O$_4$ scaling factor

The issue of the O$_4$ scaling factor is still an unresolved conundrum. MAPA results strongly depend on the choice of the SF. For CINDI–2, a SF of about 0.8 results in much better agreement to AERONET, while the unscaled O$_4$ dSCDs result in low biased AODs by a factor of 2, and a far higher number of sequences are flagged.

Thus the SF is a general limitation of MAX-DOAS analysis. As shown in Wagner et al. (2018), the discrepancies between modeled and measured $S$ can in some cases not be explained by the involved uncertainties of e.g. temperature and pressure profiles, O$_4$ cross section uncertainty, etc.

The MAPA option of determining the best matching SF (see section 2.6), allowing to analyse the dependency of the SF on various observation conditions, might help to investigate and hopefully clarify this issue in the future.

### 10 5.1.5 Flags

Profile inversions yield a best estimate for aerosol and trace gas profiles, but no direct clue on whether this profile is realistic or not. Thus, within MAPA flags have been defined based on plausibility criteria and basic uncertainty information such as the RMS of the forward model and the DOAS fit error of input dSCDs. The thresholds have been defined carefully and the sensitivity of the a-priori has been investigated in the previous section. But still, it cannot be ruled out that "good" profiles are
flagged, as well as that "bad" profiles are not yet flagged.

So far, flags have been investigated based on CINDI-2 measurements and synthetic dSCDs (see Frieß et al., in prep.). Further investigations for different instruments and measurement conditions will be made possible by the automatized processing within the FRM4DOAS project. Further extensive validation is desirable, preferably to actual profile measurements from e.g. sondes or drones.

### 20 5.2 Specific limitations of MAPA

### 5.2.1 Profile parameterization

The simple profile parameterization can only represent a limited set of profile shapes. In particular, multi-layer profiles (like a surface-near pollution plus an elevated layer) are not covered by the parameterization.

But also pure exponential profile shapes, which are often assumed in synthetic data and might be considered as "simple"
cases, are not directly included in the current MAPA parameterization. They would result from the limit of $h \to 0$ and $s \to 0$, but this limit is not covered by the dAMF LUT. Thus the MAPA results show less good agreement for exponential profiles compared to e.g. box profiles for synthetic data (Frieß et al., in prep.).

### 5.2.2 dAMF LUT

The dAMF LUT has been calculated with the MC RTM McArtim (Deutschmann et al., 2011). Thus the calculated dAMFs are
affected by MC noise. This might become relevant in case of low dAMFs which occur for low VCDs.




In addition, the dAMFs for given geometry and profile parameters is derived from the multi-dimensional dAMF LUT by linear interpolation, though the dependencies are generally nonlinear.

Based on the MAPA results for synthetic dSCDs (Frieß et al., in prep.), both effects can be considered as noncritical.

### 5.2.3 Averaging Kernels

Averaging kernels are not provided by MAPA. But still, the information on the sensitivity of MAPA for different vertical layers is inwoven in the dAMF LUTs. Further investigations will be made in the future how far the dAMF LUTs used for aerosol and trace gas inversion by MAPA might be used to construct an averaging kernel proxy.

## 6  Conclusions

The MAinz Profile Algorithm MAPA retrieves lower tropospheric profiles of aerosol extinction and trace gas concentrations from dSCD sequences derived from MAX-DOAS measurements. MAPA is based on a simple profile parameterization. In contrast to previous parameter-based profile inversion schemes, MAPA uses a MC approach to derive a distribution of best matching parameter sets (and associated profiles) rather than just one best solution. This is much faster, can deal with correlation of parameters and multiple minima, and allows to also derive an estimate of profile uncertainties. In addition, a two-stage scheme is provided for flagging probably dubious and errorneous results by warning and error, respectively, based on several criteria.

MAPA aerosol results during CINDI–2 agree well to AERONET AOD only if a scaling factor of 0.8 is applied for $O_4$, for reasons still not understood. In this context, the option of having a variable SF in MAPA might help to solve this issue in the future. Trace gas results for $NO_2$ and HCHO agree well to LP measurements. The results are robust with respect to the a-priori settings for MC and flagging.

MAPA flagging removes a large fraction, but not all scenes affected by clouds. It is thus recommended to generally apply an additional cloud flagging.

MAPA performance is affected by general MAX-DOAS limitations like a-priori assumptions in RTM like aerosol scattering properties, or the usually made assumption of horizontal homogeneity, clouds, and the uncertainty caused by the basic lack of understanding of the $O_4$ SF.

In addition, complex profiles like multiple layers, which are not adequately reflected by the chosen parameterization, cannot be retrieved.

MAPA is included in the operational processing within the FRM4DOAS project. This will allow for extensive comparisons to profiles from Optimal Estimation inversion, as well as detailed studies on the $O_4$ SF, for a variety of instruments and measurement conditions in the future.

*Competing interests.* None.





*Acknowledgements.* This study has received funding from the FRM4DOAS project under the ESA contract n°4000118181/16/I-EF.

We would like to thank the CINDI–2 and FRM4DOAS communities for valuable discussions and feedback, in particular Martina Friedrich from BIRA (Brussels, Belgium), Jan-Lukas Tirpitz and Udo Frieß from IUP (Heidelberg, Germany), and Andreas Richter from IUP (Bremen, Germany).

5 Christian Borger from MPIC (Mainz, Germany) is acknowledged for valuable comments on coding and support in bug fixing.

We thank Stefan Schmitt from IUP Heidelberg for providing Long Path DOAS measurements for $NO_2$ and HCHO.

We thank Bas Henzing for his effort in establishing and maintaining AERONET measurements at Cabauw.



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





**Table 1.** Abbreviations used in text and for indexing, sorted alphabetically.

| Abbreviation | Meaning |
| --- | --- |
| aer | Aerosol |
| AOD | Aerosol optical depth |
| bm | Best match |
| CI | confidence interval |
| CINDI | Cabauw Intercomparison of Nitrogen Dioxide Measuring Instruments |
| dAMF | differential Air mass factor |
| DL | detection limit |
| DOAS | Differential Optical Absorption Spectroscopy |
| dSCD | differential Slant column density |
| EA | Elevation angle |
| ECMWF | European Centre for Medium-Range Weather Forecasts |
| err | error |
| fm | Forward model |
| FRM4DOAS | Fiducial Reference Measurements for DOAS |
| LM | Levenberg-Marquardt |
| LT | Lower troposphere |
| LUT | Look up table |
| MAX-DOAS | Multi AXis DOAS |
| MC | Monte Carlo |
| ms | measured |
| RAA | Relative azimuth angle |
| RMS | root mean squared |
| RTM | radiative transfer model |
| sel | selected |
| SF | Scaling factor (for $O_4$) |
| SZA | Solar zenith angle |
| tg | Trace gas |
| tol | tolerance |
| tot | total |
| VCD | Vertical column density |
| wm | weighted mean |

**Table 2.** Symbols used in this study, sorted chronologically.

| Section | Symbol | Meaning |
| --- | --- | --- |
| 2.1 | $\alpha$ | EA |
| | $S$ | dSCD |
| | $V$ | VCD |
| | $A$ | dAMF |
| 2.2 | $\vartheta$ | SZA |
| | $\varphi$ | RAA |
| | $M$ | number of EAs |
| | $\boldsymbol{S}$ | dSCD sequence |
| 2.3 | $z$ | altitude coordinate |
| | $p(z)$ | vertical profile |
| | $c$ | column parameter |
| | $c_{aer} \equiv \tau$ | AOD |
| | $c_{tg} \equiv V_{tg}$ | VCD |
| | $h$ | height parameter |
| | $s$ | shape parameter |
| 2.4 | $\boldsymbol{A}$ | dAMF sequence |
| 2.5 | $R$ | RMS |
| | $\beta$ | Seed of random generator |
| | $d$ | number of MC variables |
| | $a$ | sampling per MC variable |
| | $n$ | number of random parameter sets |
| | $F$ | tolerance for $R$ compared to minimum |
| 2.6 | $f$ | $O_4$ SF |
| 2.7 | $\varepsilon$ | column uncertainty proxy |
| | $\Theta$ | Flag threshold |



**Table 3.** Default values for the Monte-Carlo based inversion algorithm for MAPA v0.98.

| Variable | Default |
|---|---|
| $\beta$ | 1 |
| $a$ | 50 |
| $d$ | 3 (aer) |
| | 2 (tg) |
| $n_{\text{tot}} = a^d$ | 125000 (aer) |
| | 2500 (tg) |
| $n_{\text{sel}}$ | 100 |
| $F$ | 1.3 |
| $c_{\text{aer}}$ range | [0.0, 5.0] |
| $h$ range | [0.02, 5.0] km |
| $s$ range | [0.2, 1.8] |

**Table 4.** Warning and error threshold default values for MAPA v0.98. The meaning of the thresholds is explained in the text. The default column uncertainty $\varepsilon$ is 0.05 for aerosols and $V_{\text{err}}$ for trace gases.

| Symbol | Description | Warning | Error |
|---|---|---|---|
| $\Theta_R$ | Upper threshold for $R$ in units of $S_{\text{err}}$ | 1 | 3 |
| $\Theta_{Rn}$ | Upper threshold $R_{\text{norm}}$ | 0.05 | 0.3 |
| $\Theta_{\text{rel}}$ | Relative column tolerance | 0.2 | 0.5 |
| $\Theta_{\text{abs}}$ | Absolute column tolerance in untis of $\varepsilon$ | 1 | 4 |
| $\Theta_{\text{DL}}$ | column detection limit in untis of $\varepsilon$ | 1 | 4 |
| $\Theta_\tau$ | Upper threshold for AOD | 2 | 3 |
| $\Theta_h$ | Upper threshold for $h$ | 3 km | 4.5 km |
| $\Theta_{\text{LT}}$ | Lower threshold for LT fraction of total column | 0.8 | 0.5 |
| $\Theta_\varphi$ | Lower threshold for RAA | 15 | nan |
| $\Theta_{\varphi,\tau}$ | Lower threshold for AOD in order to raise RAA flag | 0.5 | 3 |
| $\Theta_f$ | O$_4$ SF threshold interval (Only affects variable SF mode) | [0.6, 1.2] | [0.4, 1.4] |

**Table 5.** Variations of a-priori settings (compared to the default) and their impact on the MAPA aerosol retrieval, quantified by the number of valid sequences and the AOD comparison between MAPA and AERONET (correlation coefficient $r$ and difference $\Delta\tau$). The default settings of MAPA v0.98 with a SF of $f = 0.8$ are considered as baseline. Variations A-D refer to settings of the MC algorithm (sect. 4.1). Variations a-e refer to flag thresholds (sect. 4.2). Results for a previous MAPA release, and results for different SF are included as well. For details and discussion see text.

| Setup | Variation (Default) | #Valid | $r$ | $\Delta\tau$ |
|---|---|---|---|---|
| $f$=0.8 | - | 324 | 0.874 | $0.012 \pm 0.067$ |
| A1 | $\beta$=2 (1) | 320 | 0.882 | $0.014 \pm 0.070$ |
| A2 | $\beta$=1000 (1) | 329 | 0.876 | $0.014 \pm 0.069$ |
| B1 | $a$=20 (50) | 269 | 0.882 | $0.014 \pm 0.076$ |
| B2 | $a$=100 (50) | 342 | 0.860 | $0.026 \pm 0.088$ |
| C1 | $F$=1.1 (1.3) | 389 | 0.872 | $0.026 \pm 0.072$ |
| C2 | $F$=1.5 (1.3) | 279 | 0.908 | $0.006 \pm 0.058$ |
| D1 | $s_{\min}$=0.1 (0.2) | 311 | 0.875 | $0.019 \pm 0.071$ |
| D2 | $s_{\min}$=0.5 (0.2) | 348 | 0.848 | $0.004 \pm 0.073$ |
| D3 | $s_{\max}$=1.5 (1.8) | 330 | 0.887 | $0.018 \pm 0.067$ |
| a1 | $\Theta_R$=0.5 (1) | 324 | 0.874 | $0.012 \pm 0.067$ |
| a2 | $\Theta_R$=2 (1) | 325 | 0.874 | $0.012 \pm 0.067$ |
| a3 | $\Theta_{Rn}$=0.025 (0.05) | 238 | 0.911 | $0.022 \pm 0.064$ |
| a4 | $\Theta_{Rn}$=0.1 (0.05) | 338 | 0.874 | $0.012 \pm 0.067$ |
| b1 | $\varepsilon_\tau$=0.025 (0.05) | 311 | 0.877 | $0.011 \pm 0.067$ |
| b2 | $\varepsilon_\tau$=0.1 (0.05) | 334 | 0.876 | $0.014 \pm 0.068$ |
| c1 | $\Theta_{\text{rel}}$=0.1 (0.2) | 299 | 0.894 | $0.006 \pm 0.054$ |
| c2 | $\Theta_{\text{rel}}$=0.4 (0.2) | 340 | 0.787 | $0.022 \pm 0.094$ |
| c3 | $\Theta_{\text{abs}}$=0.5 (1) | 311 | 0.877 | $0.011 \pm 0.067$ |
| c4 | $\Theta_{\text{abs}}$=2 (1) | 334 | 0.876 | $0.014 \pm 0.068$ |
| d1 | $\Theta_h$=2 (3) km | 307 | 0.916 | $0.003 \pm 0.055$ |
| d2 | $\Theta_h$=4 (3) km | 338 | 0.783 | $0.032 \pm 0.124$ |
| e1 | $\Theta_\tau$=1 (2) | 323 | 0.874 | $0.012 \pm 0.067$ |
| e2 | $\Theta_\tau$=3 (2) | 327 | 0.874 | $0.012 \pm 0.067$ |
| v0.96 | | 337 | 0.826 | $0.037 \pm 0.126$ |
| $f$=1.0 | - | 218 | 0.905 | $-0.115 \pm 0.043$ |
| variable $f$ | - | 356 | 0.873 | $-0.018 \pm 0.069$ |





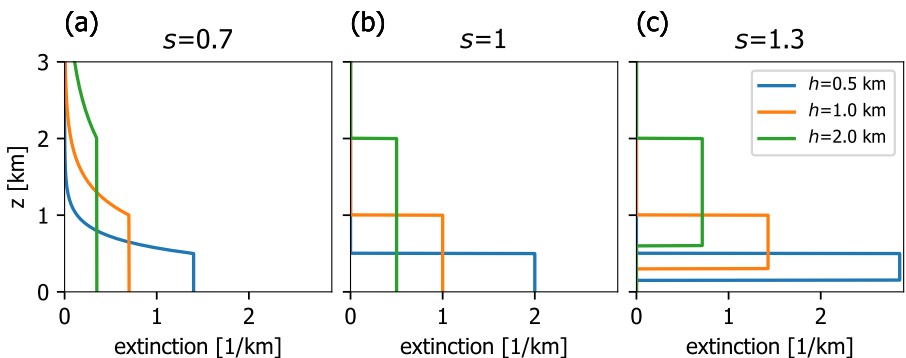

**Figure 1.** Illustration of the profile parameterization. Aerosol extinction profiles are shown for $c_{\mathrm{aer}} \equiv \tau = 1$, different heights $h$ (color coded), and shape parameters $s = 0.7$ (a), 1.0 (b), and 1.3 (c).

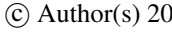



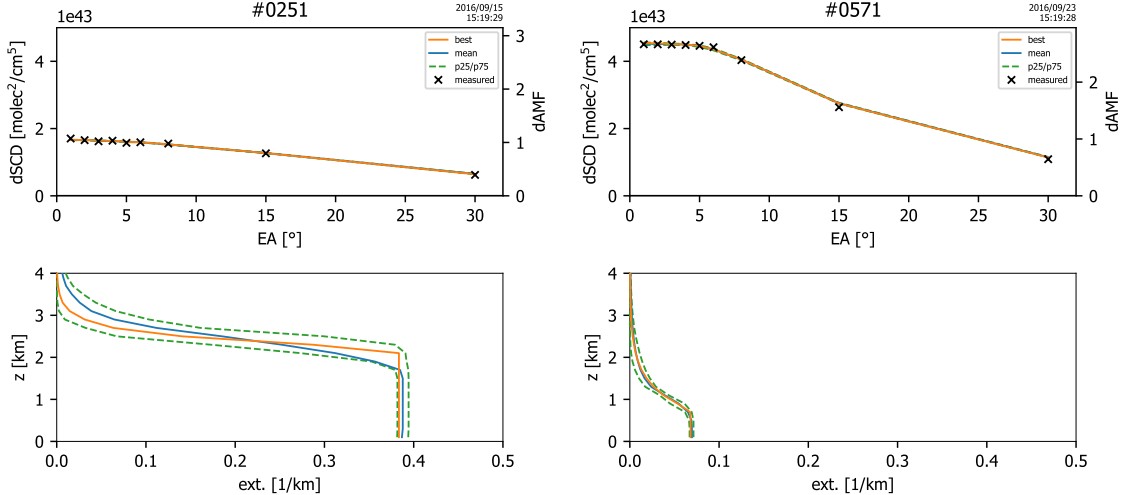

**Figure 2.** Illustration of the profile inversion for dSCD sequences of $O_4$ from 15 September (left) and 23 September (right) 2016. A scaling factor of 0.8 has been applied (see section 2.6). Top: measured and modeled dSCDs. The parameter ensembles are represented by statistical key quantities. Bottom: Corresponding vertical profiles.

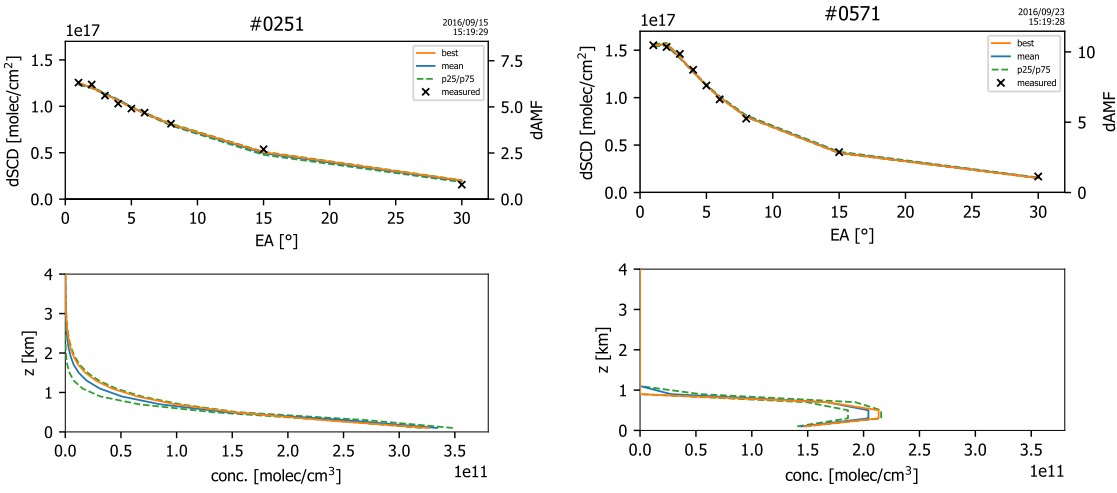

**Figure 3.** Illustration of the profile inversion for dSCD sequences of $NO_2$ from 15 September (left) and 23 September (right) 2016, based on the aerosol retrievals shown in Fig. 2. Top: measured and modeled dSCDs. The parameter ensembles are represented by statistical key quantities. Bottom: Corresponding vertical profiles.





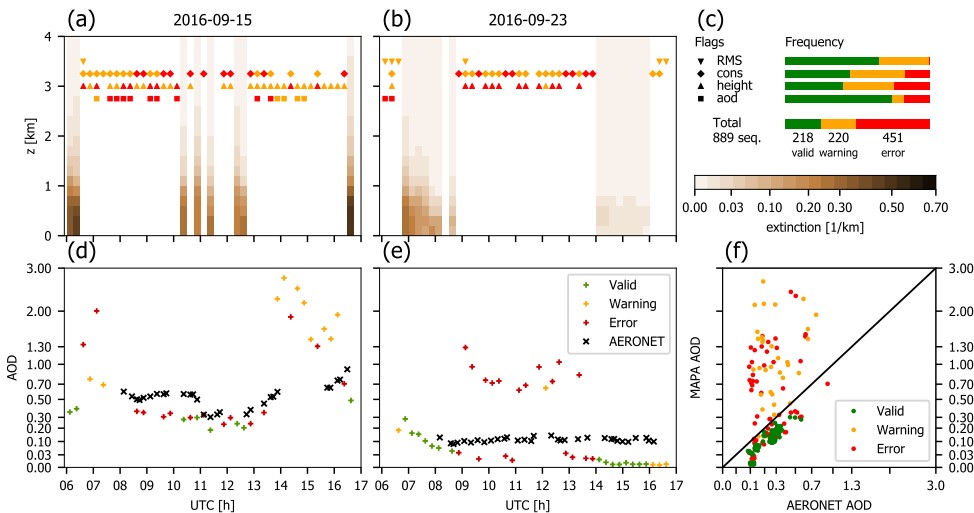

**Figure 4.** MAPA results for aerosols during CINDI-2. (a) Vertical extinction profile on 15 September. Gaps are flagged as warning (orange) or error (red), indicated by different symbols for the different flag criteria. (b) as (a) for 23 September. (c) Flag statistics for the whole CINDI-2 campaign. (d) AOD from MAPA compared to AERONET for 15 September. (e) as (d) for 23 September. (f) MAPA AOD compared to AERONET for the whole CINDI-2 campaign.





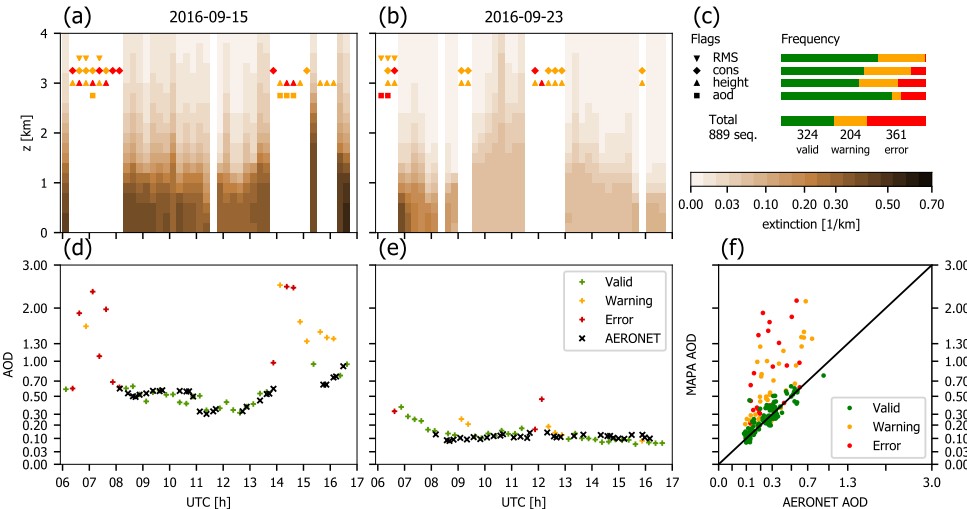

**Figure 5.** As fig. 4 but for a SF of 0.8.

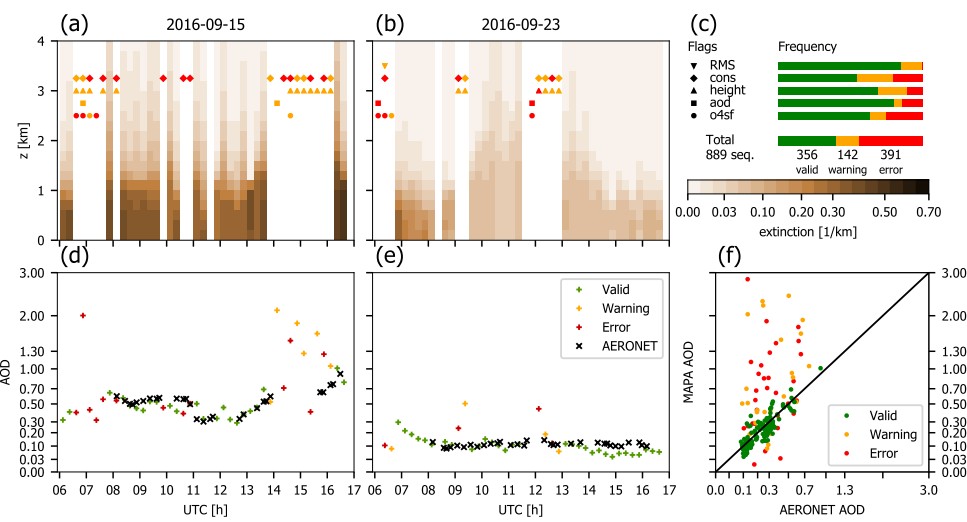

**Figure 6.** As fig. 4 but for a variable (best matching) SF.





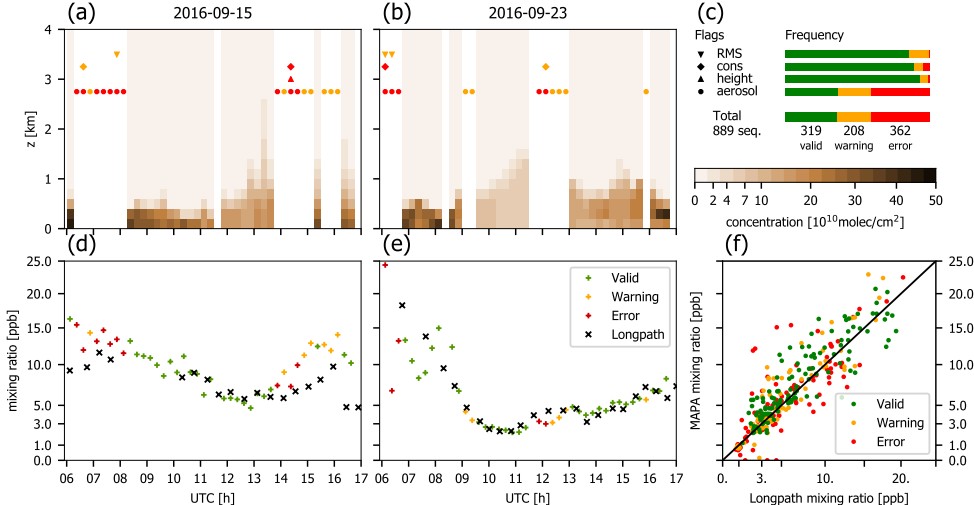

**Figure 7.** MAPA results for NO$_2$ during CINDI-2, based on aerosol profiles retrieved with a SF of 0.8. (a) Vertical extinction profile on 15 September. (b) as (a) for 23 September. (c) Flag statistics for the whole CINDI-2 campaign. (d) Mixing ratio in lowest layer (0-200m above ground) from MAPA compared to Long Path (LP) DOAS results for 15 September. (e) as (d) for 23 September. (f) MAPA lowest layer mixing ratio compared to LP for the whole CINDI-2 campaign.

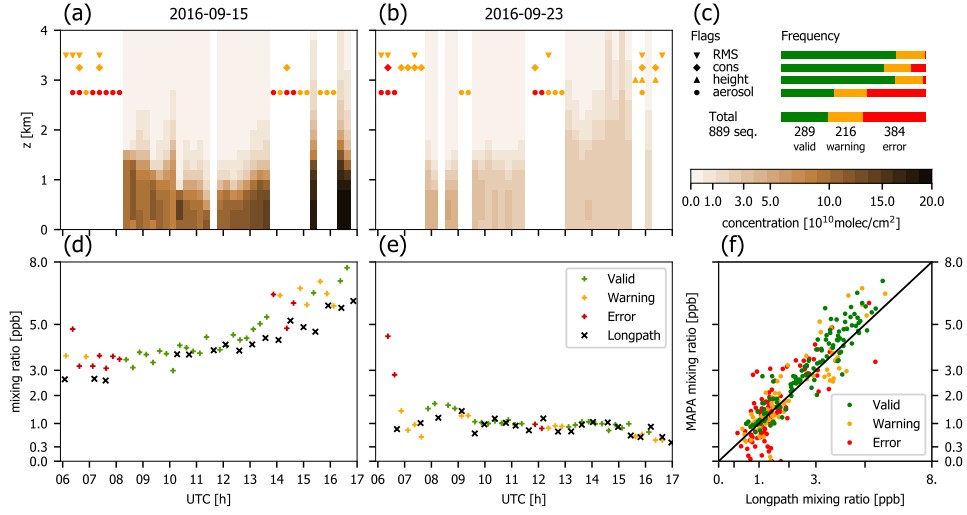

**Figure 8.** As fig. 7 but for HCHO.



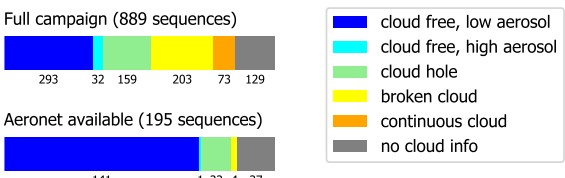

**Figure 9.** Frequence of cloud conditions as classified based on the procedure described in Wagner et al. (2016) with adjusted thresholds for CINDI-2. Missing cloud information is related to missing $O_4$ dSCDs for single elevation angles. Top: All available elevation sequences. Bottom: Only sequences where AERONET measurements are available.

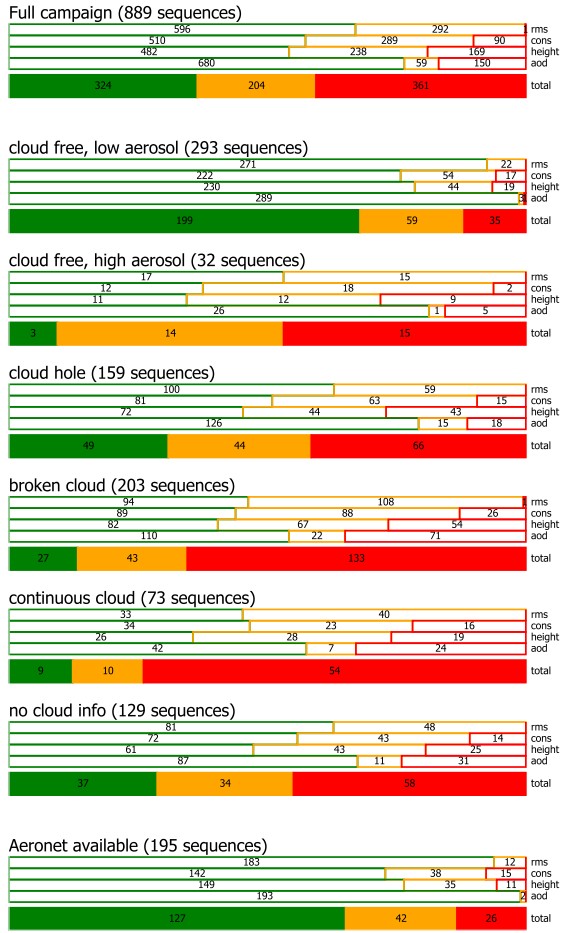

**Figure 10.** Statistics of MAPA flags for different cloud conditions.





**Appendix A: LUTs for dAMFs**

dAMFs for $O_4$ and trace gases are derived from RTM calculations using McArtim (Deutschmann et al., 2011) for a set of viewing geometries and profile parameters. The results are stored in a multi-dimensional LUT in netCDF format, which is interpolated linearly within the MAPA forward model. Table A1 lists the nodes of the parameters in the LUT. Table A2
5   provides additional settings and a-priori assumptions made for the RTM calculation. Currently, additional LUTs with other settings are calculated (starting with elevated ground altitude which will be automatically be used for elevated stations in future MAPA versions).

Note that the LUT approach used within MAPA allows for any combination of SZA and RAA, while parameter based profile retrievals shown in previous studies (Wagner et al., 2011; Frieß et al., 2016) were based on LUTs calculated only for the actual
10  SZA/RAA combinations matching the time and place of the measurements.

So far, LUTs are calculated for a set of wavelengths covering the UV and blue spectral range. For a given MAXDOAS-retrieval, MAPA v0.98 just takes the LUT with closest match in wavelength (per default: center of DOAS fit window, can be modified in configuration). In future interpolation in wavelength will also be possible.

**Table A1.** Nodes of the LUT for dAMFs. Note that other variables like wavelength, detector altitude, or aerosol settings are not included as nodes, but one LUT is determined for each combination of these additional parameters. Compare table A2.

| Variable | Symbol | unit | nodes |
|---|---|---|---|
| EA | $\alpha$ | ° | 1, 2, 3, 4, 5, 6, 8, 10, 15, 20, 30, 45, 90 |
| SZA | $\vartheta$ | ° | 10, 20, 30, 40, 50, 60, 70, 80, 85 |
| RAA | $\varphi$ | ° | 0, 5, 10, 20, 30, 60, 90, 120, 150, 180 |
| AOD | $c_{aer} \equiv \tau$ | - | 0.05, 0.1, 0.2, 0.3, 0.5, 0.7, 1.0, 1.5, 2.0, 3.0 |
| height | $h_{aer}, h_{tg}$ | km | 0.02, 0.1, 0.2, 0.3, 0.5, 0.7, 1.0, 1.2, 1.5, 1.75, 2.0, 2.5, 3.0, 5.0 |
| shape | $s_{aer}, s_{tg}$ | - | 0.1, 0.2, 0.3, 0.4, 0.5, 0.7, 1.0, 1.2, 1.5, 1.8 |

**Table A2.** RTM settings for LUT calculation. Every combination (so far: different wavelengths) is stored as separate LUT. Further LUTs for other wavelengths, ground altitudes, and aerosol settings are currently calculated and will be provided when ready.

| Variable | unit | value(s) |
|---|---|---|
| wavelength | nm | 325, 343, 360, 430, 477 |
| Single scattering albedo | - | 0.95 |
| Henye-Greenstein asymmetry parameter | - | 0.68 |
| Ground altitude (above sea level) | m | 0 |
| Detector altitude (above ground) | m | 0 |
| Ground albedo | | 0.05 |

