# Peer review of "The Mainz Profile Algorithm (MAPA)"

_Atmospheric Measurement Techniques, 2018_

## Referee Comment (RC1) · Anonymous Referee #2 · 17 Dec 2018

This paper reports a new MAX-DOAS profiling algorithm detailedly. The algorithm is based on a scientific and reasonable method. The results have good correlation with the results from the other instruments. In general the scientific topic is meaningful. Specific comments: 1, The title of this paper is about a NEW algorithm, so you should highlight what is really NEW and innovative in your algorithm, and what are the advantages comparing to the other MAX-DOAS profiling algorithms. These points should also be included in the Abstracts. 2, In the chapter about CINDI-2 campaign, the results are compared with the results from other instruments. However, it is also important to compare with the MAX-DOAS result from the same instrument but retrieved with the other algorithms. 3, In the description of the algorithm, it is better to use the symbols that are commonly used in the related papers. For example, in Equation (1), it is better to use "AMF" instead of "A", "SCD" instead of "S", and "VCD" instead of "V". In other equations, they have the same problem. 4, How accurate is the

retrieval results when the distribution of aerosol and trace gases is high (i.e. 3km). In addition, it will be better if the aerosol and trace gases profiles retrieved using MAPA are validated by corresponding profiles measured using other instruments (i.e. air balloon). Minor comments: 1, In Figure7 and 8, "mixing ratio [ppb]" => "Mixing ratio [ppb]" 2, page4 line1, "to be retrieved first as perquisite for trace gas inversions" => "to be retrieved first as a prerequisite for trace gas inversions" 3, page5 line6, "increase from ground to h" => "increase from the ground to h" 4, Page 5 Line21, "aerosol profiles, and trace gases", "comma" and "and" can't be used together. Delete comma. 5, Page 18 Line19, "cloud, and no sequence", "comma" and "and" can't be used together. Delete comma. Correct this mistake throughout your manuscript 6, Page 7 Line25, "if lowest R is" to "if the lowest R is" 7, Page 15 Line12, "we focus of variations of" to "we focus on variations of" 8, Page 18 Line30, "cloud scenes still remains" to "cloud scenes still remain" 9, Page 19 Line21, "Currently, an MAX-DOAS" to "Currently, a MAX-DOAS"

Please also note the supplement to this comment:
https://www.atmos-meas-tech-discuss.net/amt-2018-375/amt-2018-375-RC1-supplement.pdf

---

## Referee Comment (RC2) · Anonymous Referee #1 · 2 Jan 2019

**General comments**

Beirle et al. introduce the Mainz Profile Algorithm (MAPA) on the example of measurements taken during the CINDI-2 campaign. The algorithm is based on parametrization and depends on a pre-calculated LUT. The algorithm itself, its a priori assumptions, a flagging scheme, as well as the still discussed and unsolved issue of an O4 scaling factor (SF) are thoroughly discussed. The manuscript is well structured and the results show good agreement with independent measurements. However, the authors should clarify three major issues:

1. A new version of MAPA is presented but the description of differences to older versions is split up across the complete manuscript (e.g. in Sec. 1, 2.3, 2.5). Please provide one single section with differences to the older versions and

relevant improvements. Furthermore, a brief outlook of features (also new nodes for the LUT) which will be implemented in the near feature should be given. It is interesting for users to know which aerosol settings will be available soon (which SSA and asymmetry factors).

2. Figure 6 depicts results for a variable scaling factor. Unfortunately, the corresponding SF are not shown. Since these variable SF are also discussed in Section 4.4, it would be interesting to show the variability of the SF and the dependence on different flags and profiles.

3. The flagging discussion in Section 4 is questionable as specific flags are changed while keeping the other flags at their default values. As the discussion of flagging is valuable because it hasn't been covered thoroughly in other publications, this analysis should be repeated by applying and changing one flag at a time. How else could you know, if the change in one flag does not mainly affect profiles which were already flagged by other thresholds? Furthermore, it would be interesting to see the actual (AERONET) values of asymmetry factor and SSA, together with the information of the flagging scheme, to identify inaccuracies based on a wrong aerosol assumption.

**Specific comments**

**Table A1:** Why are the RAA values chosen that coarse for RAA $\geq 30°$ ? I would expect that results might change a lot for backward scattering, depending on the aerosol phase function, when changing the RAA results from e.g. 180 to 165.

**P4, L7:** You wrote that p and T profiles are extrapolated when surface values are provided. How is this extrapolation done? How large would you estimate the uncertainties when doing this extrapolation?

**P8, L6:** I would add that the agreement might be similar but it is also allowed to be slightly worse based on the definition of $R/R_{bm} < F$.

**P8, L9:** Please add here that the weighting with $1/R^2$ is referred to as weighted mean because the question about the weighting might arise in Line 19.

**P8, L25, Fig 2,3:** Thank you for changing the line width during the quick access review. However, now the min and max curves are missing.

I was wondering about these min/max curves in the first version of these figures. The curves represent aerosol scenarios with different AOTs (roughly estimated as 0.74, 1.47 on 15/09). How is it possible that these different AOTs do not lead to larger deviations in the O4 dSCD depicted in the corresponding sub-figures? Same for NO2?

**Fig 2,3:** I would suggest to change the x-axis of the EA/dSCD plots to a numbering of EA instead of the actual values. I this way, the more important details for lower EA are easier to identify when using an equidistant spacing.

**Tab 4:** Since $\Theta_R$ scales with $S_{err}$, please add information about this in Table 4.

**Fig 4-8:**

1. Please show all profiles in this kind of plots and use e.g. red rectangles around the flagged profiles to further indicate the discarded scenarios.

2. It would help if you could add regression lines (and corresponding parameters) and Pearson's correlation coefficient to the figures, for valid and valid+flagged profiles respectively.

3. Please add error bars to the sub-figures d-f.

4. Additional markers for the cloud classification scheme from Section 4.5 should be added to indicate the cloudiness during the corresponding measurement.

**Fig 4:** I am wondering why MAPA finds nearly all profiles as having issues with the

[Figure]

height flag, on 15/09. When considering that the aerosol load was mainly concentrated close to the surface (Fig 5), this indicates an issue with the algorithm or the flagging scheme/threshold. I would not expect a deviation in the profile shape when no SF is applied. And why is one warning enough to discard the corresponding profile? This appears to be a bit too strict.

**P13, L29:** Please add a time series of this variable scaling factor and a brief discussion.

**Fig 6,7:** Do you have an explanation why your results and LP DOAS data differ mostly in the morning hours (and the late afternoon)? Is this also a problem for the other days of the investigated time period?

**P16, L6:** If a lower F leads to more profiles and the correlation is not deteriorating much, I am wondering why the default is 1.1? Furthermore, when F = 1.3 leads to less profiles due to consistency issues, isn't it possible that the consistency threshold is the problem?

**P16, L27-28:** An increase of the threshold $R_n$ leads to more profiles without a deterioration in r. Could you please test if this is still true for an even larger increase?

**P17, L12:** "Here we focus of..." → "Here we focus on..."

**P17, L14, L29:** Here I do not see the point in using 3km v0.96 led to r = 0.826 and $\Theta_h = 4$ led to r = 0.783 with 337 and 338 profiles, respectively. One single profile was responsible for this drastic decrease? I would rather say that the individual scenarios at the prevalent site and time led to the conclusion of using 3km instead of 4. This might be completely different for other measurement locations, even though the sensitivity is highest for the lowest altitudes.

**P18, L6:** If the variable SF leads to a similar agreement but more profiles remain, why isn't that the preferred option?

**P20, L24:** What would be the retrieval response to an exponential scenario? Did the authors make some tests to see if some of the flagged profiles are just exponentially shaped and therefore maybe poorly retrieved by MAPA?

**P21, L1:** "...profile parameters is derived..." → "...profile parameters are derived..."

---

## Author Comment (AC1) · 25 Jan 2019

**Reply to comments from Referee #2**

We would like to thank Referee #2 for the thorough review of our manuscript and valuable feedback. Below we reply to the raised issues one by one.

*This paper reports a new MAX-DOAS profiling algorithm detailedly. The algorithm is based on a scientific and reasonable method. The results have good correlation with the results from the other instruments. In general the scientific topic is meaningful.*

*Specific comments:*

*1, The title of this paper is about a NEW algorithm, so you should highlight what is really NEW and innovative in your algorithm, and what are the advantages comparing to the other MAX-DOAS profiling algorithms. These points should also be included in the Abstracts.*

In fact the title does not claim that the paper is about a NEW algorithm. But we understand that it is not fully clear from the current manuscript what is actually new of the described MAPA algorithm. In order to clarify this issue, we extended the abstract and methods respectively:

Revised abstract:

Abstract. The Mainz profile algorithm MAPA derives vertical profiles of aerosol extinction and trace gas concentrations from MAX-DOAS measurements of slant column densities under multiple elevation angles. This manuscript presents (a) a detailed description of the MAPA algorithm (v0.98), (b) results for the CINDI-2 campaign, and (c) sensitivity studies on the impact of a-priori assumptions such as flag thresholds.

**Like previous profile retrieval schemes developed at MPIC, MAPA is based on a profile parameterization combining box profiles, which also might be lifted, and exponential profiles. But in contrast to previous inversion schemes based on least-square fits, MAPA follows a Monte Carlo approach for deriving those profile parameters yielding best match to the MAX-DOAS observations. This is much faster, and directly provides physically meaningful distributions of profile parameters. In addition, MAPA includes an elaborated flagging scheme for the identification of questionable or dubious results.**

The AODs derived with MAPA for the CINDI-2 campaign show good agreement to AERONET if a scaling factor of 0.8 is applied for $O_4$, and the respective $NO_2$ and HCHO surface mixing ratios match those derived from coincident long-path DOAS measurements. MAPA results are robust **with respect** to modifications of the a-priori MAPA settings within plausible limits.

New section:

**2.1 Heritage and advancements**

**MAPA founds on the parameterized profile inversion approach described in Li et al. (2010) or Wagner et al. (2011). It uses similar profile parameter definitions as Wagner et al. (2011) and forward models linking those parameters to dSCD sequences.**

**Main advancements of MAPA as compared to Wagner et al. (2011) are:**

- **MAPA is completely rewritten from the scratch in Python.**
- **All settings are easily adjustable by separate configuration files.**
- **MAPA provides the option of a variable scaling factor for $O_4$ (see section 2.7)**
- **MAPA uses a Monte-Carlo approach for the profile inversion (see section 2.6), while Wagner et al. (2011) used a least-squares algorithm.**
  **The MC approach is faster and provides physically meaningful uncertainty information.**
- **MAPA provides an elaborated flagging scheme for the identification of questionable results (section 2.8).**

**In the sections below we provide a full description of the MAPA profile inversion algorithm, including also parts which have been described before (like the profile parameterization) for sake of clarity and completeness.**

*2, In the chapter about CINDI-2 campaign, the results are compared with the results from other instruments. However, it is also important to compare with the MAX-DOAS result from the same instrument but retrieved with the other algorithms.*

We fully agree that comparisons with other inversion algorithms is essential. However, within this study, we focus on the description of the MAPA algorithm itself and selected results.

Within the ESA FRM4DOAS project (http://frm4doas.aeronomie.be/), extensive comparisons of different inversion schemes (both OE and parameter based) have been performed for both synthetic as well as measured dSCD sequences. The results of these studies are or will be published in near future:

- Frieß et al., Intercomparison of MAX-DOAS Vertical Profile Retrieval Algorithms: Studies using Synthetic Data, Atmos. Meas. Tech. Discuss., https://doi.org/10.5194/amt-2018-423, in review, 2018.
- Tirpitz et al., MAX-DOAS profiles for CINDI–2, in preparation.
- Richter et al., FRM4DOAS verification report, in preparation.

We have added the following sentence to the conclusions:

**Within the FRM4DOAS project, different parameter-based as well as OE-based profile inversion algorithms have been compared extensively for synthetic dSCDs (Frieß et al., 2018) as well as real measurements (Tirpitz et al., in prep.; Richter et al., in prep.).**

*3, In the description of the algorithm, it is better to use the symbols that are commonly used in the related papers. For example, in Equation (1), it is better to use "AMF" instead of "A", "SCD" instead of "S", and "VCD" instead of "V". In other equations, they have the same problem.*

We understand that abbreviations like *AMF* and *VCD* would be easier to digest. Still, we prefer single letters as symbols for variables in all equations (as recommended by NIST: https://www.nist.gov/pml/nist-guide-si-chapter-10-more-printing-and-using-symbols-and-numbers-scientific-and-technical), whereas *AMF* within an equation might be read as *AxMxF*.

Table 2 helps the reader to quickly understand the meaning of symbols/variables used in the equations throughout the document.

*4, How accurate is the retrieval results when the distribution of aerosol and trace gases is high (i.e. 3km).*

The profile parameterization used in MAPA includes the height parameter *h*. We have investigated the dependency of the ratio of AOD from MAPA versus AERONET on *h*:

[Figure]

**Figure R2-1:** Dependency of the AOD from MAPA vs. AERONET as function of the height parameter h.

This clearly demonstrates that MAPA results are not at all trustable for high *h* (interestingly, MAPA AOD is always higher than AERONET for these cases). Thus, the height parameter is used for defining the height flag, and to discard all measurements with *h*>3km. From the figure above, this criterion might even be chosen more strictly in the future.

We have included this figure to the revised manuscript in order to support the discussion of the threshold for the height flag.

*In addition, it will be better if the aerosol and trace gases profiles retrieved using MAPA are validated by corresponding profiles measured using other instruments (i.e. air balloon).*

We fully agree that accurate independent profile measurements are desirable for validation of MAX-DOAS inversion schemes. Within CINDI-2, some $NO_2$ sonde measurements have been performed by KNMI, generally revealing polluted boundary layers of about 500m altitude, in agreement with the MAPA profiles. These sonde measurements are included in the extensive CINDI-2 profiling intercomparison by Tirpitz et al. (in preparation).

*Minor comments:*

*1, In Figure7 and 8, "mixing ratio [ppb]" => "Mixing ratio [ppb]"*

Done.

*2, page4 line1, "to be retrieved first as perquisite for trace gas inversions" => "to be retrieved first as a prerequisite for trace gas inversions"*

Fixed.

*3, page5 line6, "increase from ground to h" => "increase from the ground to h"*

Fixed.

*4, Page 5 Line21, "aerosol profiles, and trace gases", "comma" and "and" can't be used together. Delete comma.*

We have modified the sentence to

Below, the forward models will be described for both $O_4$ **(**which is the basis for retrieving aerosol profiles**)** and trace gases.

*5, Page 18 Line19, "cloud, and no sequence", "comma" and "and" can't be used together. Delete comma. Correct this mistake throughout your manuscript*

5 We are not aware of a general rule that prohibits the usage of "and" after a comma. On the contrary, according to https://www.grammarly.com/blog/comma-before-and/, the usage of a comma before "and" is needed when joining two independent clauses.

We will ask the Copernicus copyeditor for guidance for the respective sentence.

*6, Page 7 Line25, "if lowest R is" to "if the lowest R is"*

We have modified the sentence to

"if lowest **RMS values** are always found for …".

*7, Page 15 Line12, "we focus of variations of" to "we focus on variations of"*

Fixed.

*8, Page 18 Line30, "cloud scenes still remains" to "cloud scenes still remain"*

Fixed.

*9, Page 19 Line21, "Currently, an MAX-DOAS" to "Currently, a MAX-DOAS"*

Fixed.

---

## Author Comment (AC2) · 25 Jan 2019

**Reply to comments from Referee #1**

We would like to thank Referee #1 for the thorough review of our manuscript and several helpful hints for improvements. Below we reply to the raised issues one by one.

*General comments*

*Beirle et al. introduce the Mainz Profile Algorithm (MAPA) on the example of measurements taken during the CINDI-2 campaign. The algorithm is based on parametrization and depends on a pre-calculated LUT. The algorithm itself, its a priori assumptions, a flagging scheme, as well as the still discussed and unsolved issue of an O4 scaling factor (SF) are thoroughly discussed. The manuscript is well structured and the results show good agreement with independent measurements. However, the authors should clarify three major issues:*

*1. A new version of MAPA is presented but the description of differences to older versions is split up across the complete manuscript (e.g. in Sec. 1, 2.3, 2.5). Please provide one single section with differences to the older versions and relevant improvements.*

We have revised the abstract, clarifying which parts of MAPA are actually new. In addition, we have added a new method section 2.1 on "Heritage and advancements", pointing out the heritage of previous profile inversion algorithms (parameterization, LUT based) versus the new developments within MAPA v0.98 (completely new python implementation, MC approach for determination of best matching profiles and uncertainties, extensive flagging scheme).

Revised abstract:

Abstract. The Mainz profile algorithm MAPA derives vertical profiles of aerosol extinction and trace gas concentrations from MAX-DOAS measurements of slant column densities under multiple elevation angles. This manuscript presents (a) a detailed description of the MAPA algorithm (v0.98), (b) results for the CINDI-2 campaign, and (c) sensitivity studies on the impact of a-priori assumptions such as flag thresholds.

**Like previous profile retrieval schemes developed at MPIC, MAPA is based on a profile parameterization combining box profiles, which also might be lifted, and exponential profiles. But in contrast to previous inversion schemes based on least-square fits, MAPA follows a Monte Carlo approach for deriving those profile parameters yielding best match to the MAX-DOAS observations. This is much faster, and directly provides physically meaningful distributions of profile parameters. In addition, MAPA includes an elaborated flagging scheme for the identification of questionable or dubious results.**

The AODs derived with MAPA for the CINDI-2 campaign show good agreement to AERONET if a scaling factor of 0.8 is applied for $O_4$, and the respective $NO_2$ and HCHO surface mixing ratios match those derived from coincident long-path DOAS measurements. MAPA results are robust **with respect** to modifications of the a-priori MAPA settings within plausible limits.

New section:

**2.1 Heritage and advancements**

**MAPA founds on the parameterized profile inversion approach described in Li et al. (2010) or Wagner et al. (2011). It uses similar profile parameter definitions as Wagner et al. (2011) and forward models linking those parameters to dSCD sequences.**

**Main advancements of MAPA as compared to Wagner et al. (2011) are:**

- **MAPA is completely rewritten from the scratch in Python.**
- **All settings are easily adjustable by separate configuration files.**
- **MAPA provides the option of a variable scaling factor for $O_4$ (see section 2.7)**
- **MAPA uses a Monte-Carlo approach for the profile inversion (see section 2.6), while Wagner et al. (2011) used a least-squares algorithm.**
  **The MC approach is faster and provides physically meaningful uncertainty information.**
- **MAPA provides an elaborated flagging scheme for the identification of questionable results (section 2.8).**

**In the sections below we provide a full description of the MAPA profile inversion algorithm, including also parts which have been described before (like the profile parameterization) for sake of clarity and completeness.**

*Furthermore, a brief outlook of features (also new nodes for the LUT) which will be implemented in the near feature should be given. It is interesting for users to know which aerosol settings will be available soon (which SSA and asymmetry factors).*

We have updated Appendix A to the current state of available LUTs. In addition, we provide MAPA LUTs at [ftp://ftp.mpic.de/MAPA/LUTs](ftp://ftp.mpic.de/MAPA/LUTs), and a general MAPA documentation on [ftp://ftp.mpic.de/MAPA/documentation/index.html](ftp://ftp.mpic.de/MAPA/documentation/index.html). Additional or extended LUTs will be included there as soon as available.

*2. Figure 6 depicts results for a variable scaling factor. Unfortunately, the corresponding SF are not shown. Since these variable SF are also discussed in Section 4.4, it would be interesting to show the variability of the SF and the dependence on different flags and profiles.*

We have added the variable scaling factor to Figure 6 d+e (see the reply to the comment on Fig. 4-8 below and Figure R1-2). In addition, we modified the last paragraph of section 3.1 to

**Having the option of a variable (best matching) scaling factor is a new feature of MAPA, to our knowledge not provided by any other MAX-DOAS inversion scheme. However, this additional degree of freedom adds complexity, and different effects (like aerosol properties being different from the RTM a-priori, or cloud effects) might be "tuned" to an acceptable match via the scaling factor. As the variable scaling factor has not yet been tested extensively, we focus on the results for a fixed SF of 0.8 as a more "familiar" and transparent setup below, but plan to systematically investigate the results of best matching SFs for various locations and measurement conditions in the near future.**

*3. The flagging discussion in Section 4 is questionable as specific flags are changed while keeping the other flags at their default values. As the discussion of flagging is valuable because it hasn't been covered thoroughly in other publications, this analysis should be repeated by applying and changing one flag at a time. How else could you know, if the change in one flag does not mainly affect profiles which were already flagged by other thresholds?*

Section 4 provides an extensive sensitivity analysis on the impact of a-priori settings. All relevant configuration settings are modified one by one within a plausible range, and the impact on those modifications on MAPA results is judged based on the agreement to AERONET measurements in comparison to the base run performance. We consider this as a reasonable end-to-end analysis which provides information on the crucial parameters and key sensitivities.

This approach exactly refers to the raised question: if we, for instance, change the height flag threshold, we see the impact on the final result, ignoring those cases already flagged by other flags.

*Furthermore, it would be interesting to see the actual (AERONET) values of asymmetry factor and SSA, together with the information of the flagging scheme, to identify inaccuracies based on a wrong aerosol assumption.*

Further investigations on the impact of aerosol properties like asymmetry and SSA would indeed be interesting. Unfortunately, the number of AERONET aerosol inversion datasets during CINDI2 is very small, such that a systematic investigation of these effects is not possible within this study.

*Specific comments*

*Table A1: Why are the RAA values chosen that coarse for RAA ≥ 30° ? I would expect that results might change a lot for backward scattering, depending on the aerosol phase function, when changing the RAA results from e.g. 180 to 165.*

We agree that the RAA nodes in the AMF LUTs should be better resolved, and we will include additional nodes in future RTM calculations. We have added a respective note to Appendix A.

*P4, L7: You wrote that p and T profiles are extrapolated when surface values are provided. How is this extrapolation done? How large would you estimate the uncertainties when doing this extrapolation?*

We have extended the respective sentence to

If ground measurements at the station are available only, they are used to construct extrapolated profiles **based on a constant lapse rate up to 12 km, and a constant temperature above (see Wagner et al., 2018, section 4.1.1, for details).**

The uncertainty of the resulting $O_4$ VCD is less than 4% (as derived from a comparison between extrapolated profiles to ECWMF).

*P8, L6: I would add that the agreement might be similar but it is also allowed to be slightly worse based on the definition of $R/R_{bm} < F$.*

We modified the sentence to

… yields an ensemble of parameter sets with **$R < F x R_{bm}$, i.e.** similar **(slightly worse)** agreement between measurement and forward model.

*P8, L9: Please add here that the weighting with $1/R^2$ is referred to as weighted mean because the question about the weighting might arise in Line 19.*

We changed the sentence to

**- weighted** mean **(wm)** and standard deviation**, with $1/R^2$ as weights**

*P8, L25, Fig 2,3: Thank you for changing the line width during the quick access review. However, now the min and max curves are missing. I was wondering about these min/max curves in the first version of these figures.*

*The curves represent aerosol scenarios with different AOTs (roughly estimated as 0.74, 1.47 on 15/09). How is it possible that these different AOTs do not lead to larger deviations in the O4 dSCD depicted in the corresponding sub-figures? Same for NO2?*

In the initial submission of this manuscript, Figures 2&3 included curves for the absolute minima/maxima extinction calculated independently for each height level. It has to be noted, however, that these curves (as well as the percentiles) do not correspond to any actual profile

within the ensemble. Thus, the min/max curves must not be interpreted as aerosol scenario, and their integral does not correspond to the respective min/max AODs.

But still the point made by the reviewer generally holds: MAPA results can in fact reveal a high variation of column parameters (of factor 4 and more). I.e., for some scenarios, the forward model finds quite similar agreement to the measured dSCDs for completely different column parameters. However, these cases are flagged by the consistency flag.

In the revised manuscript, we add the following note to the caption of Fig. 2:

**Note that the percentiles of vertical profiles are calculated independently for each height level. I.e. they do not correspond to an actual profile from the ensemble, but indicate the general level of uncertainty of vertical profiles.**

*Fig 2,3: I would suggest to change the x-axis of the EA/dSCD plots to a numbering of EA instead of the actual values. I this way, the more important details for lower EA are easier to identify when using an equidistant spacing.*

As the sequence of EAs might be different for other instruments/campaigns, we have decided not to use the (somehow arbitrary) ordinal as abscissa, but to keep the EA value itself. However, in order to emphasize the details for low EAs, we now use a nonlinear scale for EAs in the revised manuscript:

[Figure]

Figure R1-1: Updated layout of the dSCD dependency on EA, using a nonlinear scale for EA.

*Tab 4: Since $\Theta_R$ scales with $S_{err}$, please add information about this in Table 4.*

$S_{err}$ is the median DOAS fit error, as explained in 2.7.1. For clarity, we now clearly define this quantity already in section 2.2.2 (elevation sequence) and add it to Table 2.

*Fig 4-8:*

We thank the reviewer for the several suggestions of extensions to Figures 4-8. We have implemented all proposed modifications (including the results for the variable SF as raised in general comment 2), resulting in the following figure for aerosol results for variable SF (corresponding to Fig. 6 of the AMTD manuscript):

[Figure]

Figure R1-2:.Modified version of Fig. 6, including 1. flagged profiles in (a) and (b),
2. regression lines and correlation coefficients in (f), 3. error bars in (d) and (e),
4. results of the cloud classification in (a) and (b), and 5. the best matching scaling factor in (d) and (e).

We consider this figure to be too overcrowded with information, and to partly distract the reader's attention from the relevant messages. Thus, we only partly follow the reviewer's suggestions for figure modifications in the revised manuscript. Below we reply to the individual proposals, and discuss which changes are adopted in the final figure:

*1. Please show all profiles in this kind of plots and use e.g. red rectangles around the flagged profiles to further indicate the discarded scenarios.*

If all profiles are included to subplots (a) and (b), the eye is inevitably drawn to the extreme outliers. We consider the flagged profiles to be distractive without providing relevant information. Thus we decided to not include the flagged profiles in the final figures 4-8.

*2. It would help if you could add regression lines (and corresponding parameters) and Pearson's correlation coefficient to the figures, for valid and valid+flagged profiles respectively.*

We have added regression lines and correlation coefficients to subplot (f) as proposed by the reviewer.

*3. Please add error bars to the sub-figures d-f.*

Figures d-f display the best value for the fitted column parameter $c$ (aerosols) or the lowest layer concentration (trace gases). In Fig. R1-2 we have added the range of ensemble values based on the 25/75 percentiles. These ranges can be quite large for the flagged sequences. For valid sequences, however, these ranges are always small (otherwise, a consistency flag would be raised).

We consider the error bars to be distractive without adding relevant information and have decided not to include them in the revised manuscript.

*4. Additional markers for the cloud classification scheme from Section 4.5 should be added to indicate the cloudiness during the corresponding measurement.*

We have added cloud classification results to the top of subfigures (a) and (b). In the revised manuscript, we moved Fig. 9 to section 2.7.5. Thus, the legend of cloud classification is now already introduced before Figures 4-8 (becoming 5-9 in the revised manuscript).

*Fig 4: I am wondering why MAPA finds nearly all profiles as having issues with the height flag, on 15/09. When considering that the aerosol load was mainly concentrated close to the surface (Fig 5), this indicates an issue with the algorithm or the flagging scheme/threshold. I would not expect a deviation in the profile shape when no SF is applied.*

For the investigated MAPA results for CINDI2, this is indeed a clear finding: the resulting profiles are often close to the surface for a scaling factor of 0.8, while the best matching profiles without a SF generally yield higher height parameters, and are thus often discarded by the height flag, often in addition with the consistency flag.

*And why is one warning enough to discard the corresponding profile? This appears to be a bit too strict.*

For MAPA flagging, we indeed follow a quite conservative approach and decided to raise a total warning already when a single warning occurs.

The comparison studies done within FRM4DOAS (e.g., Frieß et al., 2018) also conclude that the current MAPA flagging seems to be too strict. However, by this strict flagging we nearly exclude all outliers, which are still present in the results from other algorithms (compare e.g. Fig. 16 in Frieß et al., 2018).

The comparison to AERONET and LP-DOAS indicates that the aerosol flags are generally plausible, but trace gas flags are indeed too strict. As trace gas flags are dominated by the total aerosol flag, we will check under which circumstances an aerosol warning might be acceptable within the trace gas retrievals in a future study. We have added the following statement to the conclusions:

**The MAPA flagging scheme generally succeeds in identifying dubious results, but a considerable fraction of elevation sequences is flagged. For trace gas profiles, the flagging scheme is dominated by the aerosol flag, which seems to be too strict. It has to be checked under which circumstances an aerosol warning might be acceptable within the trace gas retrievals in a future study.**

*P13, L29: Please add a time series of this variable scaling factor and a brief discussion.*

We have added the variable scaling factor to Fig. 6 (becoming Fig. 7 in the revised manuscript), and slightly extended the discussion of the SF as specified in our reply to general comment 2. An in-depth analysis of the variable SF for various conditions will be focus of a future study.

*Fig 6,7: Do you have an explanation why your results and LP DOAS data differ mostly in the morning hours (and the late afternoon)? Is this also a problem for the other days of the investigated time period?*

We have investigated the ratio of MAPA to LP DOAS as function of time of day:

[Figure]

Figure R1-3:.Ratio of lowest layer mixing ratio from MAPA vs. LP-DOAS
as function of time of day for HCHO (left) and NO$_2$ (right).

For formaldehyde, we see no indication for a time-of-day dependency. For NO$_2$, largest deviations between valid MAPA results and LP-DOAS occur between 10-12, but statistics are rather poor. We have no explanation for this finding, but will keep it in mind and investigate the diurnal cycle for other locations as well.

*P16, L6: If a lower F leads to more profiles and the correlation is not deteriorating much, I am wondering why the default is 1.1? Furthermore, when F = 1.3 leads to less profiles due to consistency issues, isn't it possible that the consistency threshold is the problem?*

We assume that the reviewer meant to ask *why the default is not 1.1*.

Admittedly, the choice of *F* is somehow arbitrary. Within the MAPA algorithm and flagging scheme, *F* basically fulfils two tasks:

- it allows to retrieve a profile ensemble rather than a single profile, thereby providing uncertainty information

- it allows to check how consistent the ensemble profiles are, thereby providing information about how sensitive the dSCDs are on profile parameters.

It is thus clear that a lower *F* results in more valid sequences, as the consistency flag is deactivated. But we don't see the consistency threshold as "problem", but as a very helpful indicator on which profiles are trustable and which not.

We extended section 4.1. C by the following statement:

**For MAPA v0.98 default settings, we stick to the choice of *F*=1.3. But we recommend to also test smaller values for *F* like 1.2 or 1.1, in particular if a large fraction of sequences is flagged by the consistency flag.**

*P16, L27-28: An increase of the threshold R$_n$ leads to more profiles without a deterioration in r. Could you please test if this is still true for an even larger increase?*

We have tested an even higher threshold for $R_n$. For the investigated CINDI-2 results, this has no effect at all on the resulting AOD and total flag statistics, simply because all effected sequences with $R_n>0.1$ are already flagged by other criteria (height, consistency, and/or AOD).

*P17, L12: "Here we focus of..." → "Here we focus on..."*

Fixed.

*P17, L14, L29: Here I do not see the point in using 3km v0.96 led to r = 0.826 and Θ h = 4 led to r = 0.783 with 337 and 338 profiles, respectively. One single profile was responsible for this drastic decrease? I would rather say that the individual scenarios at the prevalent site and time led to the conclusion of using 3km instead of 4. This might be completely different for other measurement locations, even though the sensitivity is highest for the lowest altitudes.*

As stated in section 4.3, version 0.96 uses slightly different flag definitions (not only thresholds) than version 0.98. Thus, v0.96 and variation d2 correspond to different ensembles and do not differ by just one profile.

The decreasing sensitivity with increasing height is a general restriction of the MAX-DOAS method. We have investigated the dependency of the ratio of MAPA vs. AERONET AOD on the fitted height parameter, and found a clear increase of the error:

[Figure]

**Figure R1-4:** Dependency of the AOD from MAPA vs. AERONET as function of the height parameter $h$.

This clearly demonstrates that MAPA results are not at all trustable for high $h$ (interestingly, MAPA AOD is always higher than AERONET for these cases). Thus, the height parameter is used for defining the height flag, and to discard all measurements with $h>3$km. From the figure above, this criterion might even be chosen more strictly in the future.

We include this figure to the revised manuscript in order to support the discussion of the threshold for the height flag.

Still, MAPA provides all results also for flagged scenarios, and the user is free to modify the threshold for $h$ in the configuration file.

*P18, L6: If the variable SF leads to a similar agreement but more profiles remain, why isn't that the preferred option?*

As stated above, thorough investigations of the results for variable SF will be performed as next step for various instruments and measurement conditions.

*P20, L24: What would be the retrieval response to an exponential scenario? Did the authors make some tests to see if some of the flagged profiles are just exponentially shaped and therefore maybe poorly retrieved by MAPA?*

Exponential profiles are included in the comparison study based on synthetic profiles by Frieß et al. (2018). We modified the sentence to

**Thus, for synthetic dSCDs based on exponential profiles, the MAPA results try to mimic the exponential shape by a low height parameter and low shape parameter, but performance (in terms of number of valid profiles as well as the agreement of the resulting column parameter) is worse than e.g. for box profiles (see Figures 12 and 16 in Frieß et al., 2018).**

With respect to the true profiles during CINDI-2, we cannot check if they are exponentially shaped.

*P21, L1: "...profile parameters is derived..." → "...profile parameters are derived..."*

Fixed.